# The computational nature of memory modification

**Samuel J Gershman[1]\*, Marie-H Monfils[2], Kenneth A Norman[3], Yael Niv[3]**

[1]Department of Psychology and Center for Brain Science, Harvard University, Cambridge, United States; [2]Department of Psychology, University of Texas, Austin, United States; [3]Princeton Neuroscience Institute and Department of Psychology, Princeton University, Princeton, United States

**Abstract** Retrieving a memory can modify its influence on subsequent behavior. We develop a computational theory of memory modification, according to which modification of a memory trace occurs through classical associative learning, but which memory trace is eligible for modification depends on a structure learning mechanism that discovers the units of association by segmenting the stream of experience into statistically distinct clusters (latent causes). New memories are formed when the structure learning mechanism infers that a new latent cause underlies current sensory observations. By the same token, old memories are modified when old and new sensory observations are inferred to have been generated by the same latent cause. We derive this framework from probabilistic principles, and present a computational implementation. Simulations demonstrate that our model can reproduce the major experimental findings from studies of memory modification in the Pavlovian conditioning literature.

\*For correspondence: gershman@fas.harvard.edu

## Introduction

In both humans and animals, memory retrieval is a significant learning event (*Dudai, 2012*; *Roediger and Butler, 2011*; *Spear, 1973*). A memory's strength and content can be modified immediately after retrieval, and this malleability is often more potent than new learning without retrieval. While this phenomenon is well-documented, its underlying mechanisms remain obscure. Does retrieval render the contents of the memory trace modifiable, does it affect the future accessibility of the memory trace, or both?

We develop a computational theory that begins to address these questions. Central to our theory is the idea that memory is inferential in nature: Decisions about when to modify an old memory or form a new memory are guided by inferences about the *latent causes* of sensory data (*Gershman et al., 2010*, *2014*, *2015*). Memories contain statistical information about inferred latent causes (when they are likely to occur, what sensory data they tend to generate). These statistics are retrieved and updated whenever a previously inferred latent cause is believed to have generated new sensory data. Conditions that promote the retrieval of a memory are, according to this account, precisely the conditions that promote the inference that the same previously inferred latent cause is once again active. If no previously inferred latent cause adequately predicts the current sensory data, then a new memory is formed. Thus, memory modification is intimately connected to the process of latent structure learning. We formalize this idea as a probabilistic model, and then demonstrate its explanatory power by simulating a wide range of post-retrieval memory modification phenomena.

It is important to clarify at the outset that our theory is formulated at an abstract, cognitive level of analysis, in order to elucidate the design principles and algorithmic structure of memory. We do not make strong claims about biologically plausible implementation in realistic neurons, although we

**eLife digest** Our memories contain our expectations about the world that we can retrieve to make predictions about the future. For example, most people would expect a chocolate bar to taste good, because they have previously learned to associate chocolate with pleasure. When a surprising event occurs, such as tasting an unpalatable chocolate bar, the brain therefore faces a dilemma. Should it update the existing memory and overwrite the association between chocolate and pleasure? Or should it create an additional memory? In the latter case, the brain would form a new association between chocolate and displeasure that competes with, but does not overwrite, the original one between chocolate and pleasure.

Previous studies have shown that surprising events tend to create new memories unless the existing memory is briefly reactivated before the surprising event occurs. In other words, retrieving old memories makes them more malleable. Gershman et al. have now developed a computational model for how the brain decides whether to update an old memory or create a new one. The idea at the heart of the model is that the brain will attempt to infer what caused the surprising event. The reason the chocolate bar tastes unpalatable, for example, might be because it was old and had spoiled. Every time the brain infers a new possible cause for a surprising event, it will create an additional memory to store this new set of expectations. In the future we will know that spoiled chocolate bars taste bad.

However, if the brain cannot infer a new cause for the surprising event – because, for example, there appears to be nothing unusual about the unpalatable chocolate bar – it will instead opt to update the existing memory. The next time we buy a chocolate bar, we will have slightly lower expectations about how good it will taste. The dilemma of whether to update an existing memory or create a new one thus boils down to the question: is the surprising event the consequence of a new cause or an old one? This theory implies that retrieving a memory nudges the brain to infer that its associated cause is once again active and, since this is an old cause, it means that the memory will be eligible for updating.

Many experiments have been performed on the topic of modifying memories, but this is the first computational model that offers a unifying explanation for the results. The next step is to work out how to apply the model, which is phrased in abstract terms, to networks of neurons that are more biologically realistic.

speculate about such an implementation in the Discussion. Addressing this question is a logical next step for this line of research.

## Retrieval-induced memory modification in pavlovian conditioning

While retrieval-induced memory modification has been documented in a variety of domains—including procedural (*Censor et al., 2010*; *Walker et al., 2003*), episodic (*Hupbach et al., 2007*; *Karpicke and Roediger, 2008*), and instrumental (*Lee et al., 2006b*; *Xue et al., 2012*) learning—we focus on Pavlovian conditioning, because it offers some of the most elementary and well-studied examples. During the acquisition phase of a typical Pavlovian conditioning experiment, a motivationally neutral conditional stimulus (CS; e.g., tone) is repeatedly paired with a motivationally reinforcing unconditional stimulus (US; e.g., a shock). This repeated pairing results in the animal producing a conditioned response (CR; e.g., freezing) to the CS. In a subsequent extinction phase, the CS is presented alone, and the animal gradually ceases to produce the CR. A final test phase, after some delay, probes the animal's long-term memory of the CS-US relationship by presenting the CS alone.

In a classic experiment using a Pavlovian fear conditioning task, *Misanin et al. (1968)* found that electroconvulsive shock had no effect on a fear memory acquired a day previously; however, if the animal was briefly reexposed to the acquisition cue prior to electroconvulsive shock, the animal subsequently exhibited loss of fear. This finding was followed by numerous similar demonstrations of post-retrieval memory modification (see *Riccio et al., 2006*, for a historical overview).

Contemporary neuroscientific interest in this phenomenon was ignited by *Nader et al. (2000)*, who showed that retrograde amnesia for an acquired fear memory could be produced by injection

of a protein synthesis inhibitor (PSI) into the lateral nucleus of the amygdala shortly after reexposure to the acquisition cue. Subsequent studies have provided a detailed neural and behavioral characterization of post-retrieval memory modification, describing a large cast of molecular mechanisms (*Tronson and Taylor, 2007*) and several boundary conditions on its occurrence (*Dudai, 2012*; *Duvarci and Nader, 2004*; *Nader and Hardt, 2009*). For instance, it has been shown that stronger and older memories are harder to modify following retrieval (*Suzuki et al., 2004*), and that the modification is cue-specific (*Doyère et al., 2007*).

Importantly, there is now evidence that memory modification can be obtained with a purely behavioral procedure. In particular, *Monfils et al. (2009)* and *Schiller et al. (2010)* showed, in rats and in humans, that reexposing a subject to the cue shortly (10 min to 1 hr) before extinction training is sufficient to reduce conditioned responding at test. This finding presents a deep puzzle for associative learning theory, since the cue reexposure is operationally an extinction trial and hence it is unclear what makes it special. One of our main goals will be to unravel this puzzle, showing how cue reexposure influences probabilistic beliefs about latent causes such that they are eligible for updating by the subsequent extinction training.

This body of work has traditionally been understood as probing mechanisms of 'reconsolidation' (*Nader et al., 2000*; *Przybyslawski and Sara, 1997*), under the hypothesis that memory retrieval renders the memory trace unstable, setting in motion a protein-synthesis-dependent process of synaptic stabilization. This process is thought to resemble initial post-learning consolidation, whereby a newly encoded memory gradually becomes resistant to disruption. However, this terminology is heavily theory-laden, and the explanatory adequacy of both consolidation and reconsolidation have been repeatedly questioned (*Ecker et al., 2015*; *Miller and Springer, 1973*, *Miller and Matzel, 2006*). We therefore avoid using this terminology to refer to empirical phenomena, favoring instead the less tendentious 'post-retrieval memory modification.' The relationship of our work to consolidation and reconsolidation will be revisited in the Discussion.

Before addressing the key memory modification phenomena, we first situate them within a larger set of issues in Pavlovian conditioning. These problems provide the starting point for our new theory.

## Memory and associative learning theory

Classical theories of associative learning, such as the Rescorla-Wagner model (*Rescorla and Wagner, 1972*), posit that over the course of acquisition in a Pavlovian conditioning experiment, the animal learns an association between the CS and the US, and the magnitude of the CR reflects the strength of this association. The main weakness of the Rescorla-Wagner model, and many similar models (*Pearce and Bouton, 2001*), is its prediction that presenting the CS repeatedly by itself ('extinction') will erase the CS-US association formed during acquisition—in other words, the model predicts that *extinction is unlearning*. It is widely accepted that this assumption, shared by a large class of models, is wrong (*Delamater, 2004*; *Dunsmoor et al., 2015*; *Gallistel, 2012*).

*Bouton (2004)* reviewed a range of conditioning phenomena in which putatively extinguished associations are recovered in a post-extinction test phase. For example, simply increasing the time between extinction and test is sufficient to increase responding to the extinguished CS, a phenomenon known as *spontaneous recovery* (*Pavlov, 1927*; *Rescorla, 2004*). Another example is *reinstatement*: reexposure to the US alone prior to test increases conditioned responding to the CS (*Bouton and Bolles, 1979b*; *Pavlov, 1927*; *Rescorla and Heth, 1975*). Conditioned responding can also be recovered if the animal is returned to the acquisition context, a phenomenon known as *renewal* (*Bouton and Bolles, 1979a*).

*Bouton (1993)* interpreted the attenuation of responding after extinction in terms of the formation of an extinction memory that competes for retrieval with the acquisition memory; this retroactive interference can be relieved by a change in temporal context or the presence of retrieval cues, thereby leading to recovery of the original CS (see also *Miller and Laborda, 2011*). Central to retrieval-based accounts of conditioning is the idea that the associations learned during acquisition are linked to the spatiotemporal context of the acquisition session, and as a result they are largely unaffected by extinction. Likewise, extinction results in learning that is linked to the spatiotemporal context of the extinction session. The manipulations reviewed above are hypothesized to either reinstate elements of the acquisition context (e.g., renewal, reinstatement) or attenuate elements of the extinction context (e.g., spontaneous recovery). These modifications of contextual elements

effectively change the accessibility of particular associative memories. According to this view, modification of the acquisition memory (in particular, its accessibility) should occur when the acquisition and extinction phases are linked to the same spatiotemporal context.

The major stumbling block is that it is unclear what should constitute a spatiotemporal context: What are its constitutive elements, under what conditions are they invoked, and when should new elements come into play? Existing theories have operationalized context in several (not mutually exclusive) ways: as observable stimuli [e.g., the conditioning box; *Bouton, 1993*], recent stimulus and response history (*Capaldi, 1994*), or a random flux of stimulus elements (*Estes, 1950*, *1955*). However, no computational implementation has been shown to capture the full range of memory modification phenomena that we discuss below.

## Results

### A latent cause theory

In this section, we develop a latent cause theory of Pavlovian conditioning that treats context (operationalized as the history of sensory data) as the *input* into a structure learning system, which outputs a parse of experience into latent causes—hypothetical entities in the environment that govern the distribution of stimulus configurations (*Courville, 2006*; *Courville et al., 2006*; *Gershman et al., 2010*; *Gershman and Niv, 2012*; *Gershman et al., 2013a, 2015*; *Soto et al., 2014*). Like the Rescorla-Wagner model (*Figure 1A*), our theory posits the learning of CS-US associations, but these associations are modulated by the animal's probabilistic beliefs about latent causes. New causes are inferred when existing causes fail to accurately predict the currently observed CS-US contingency

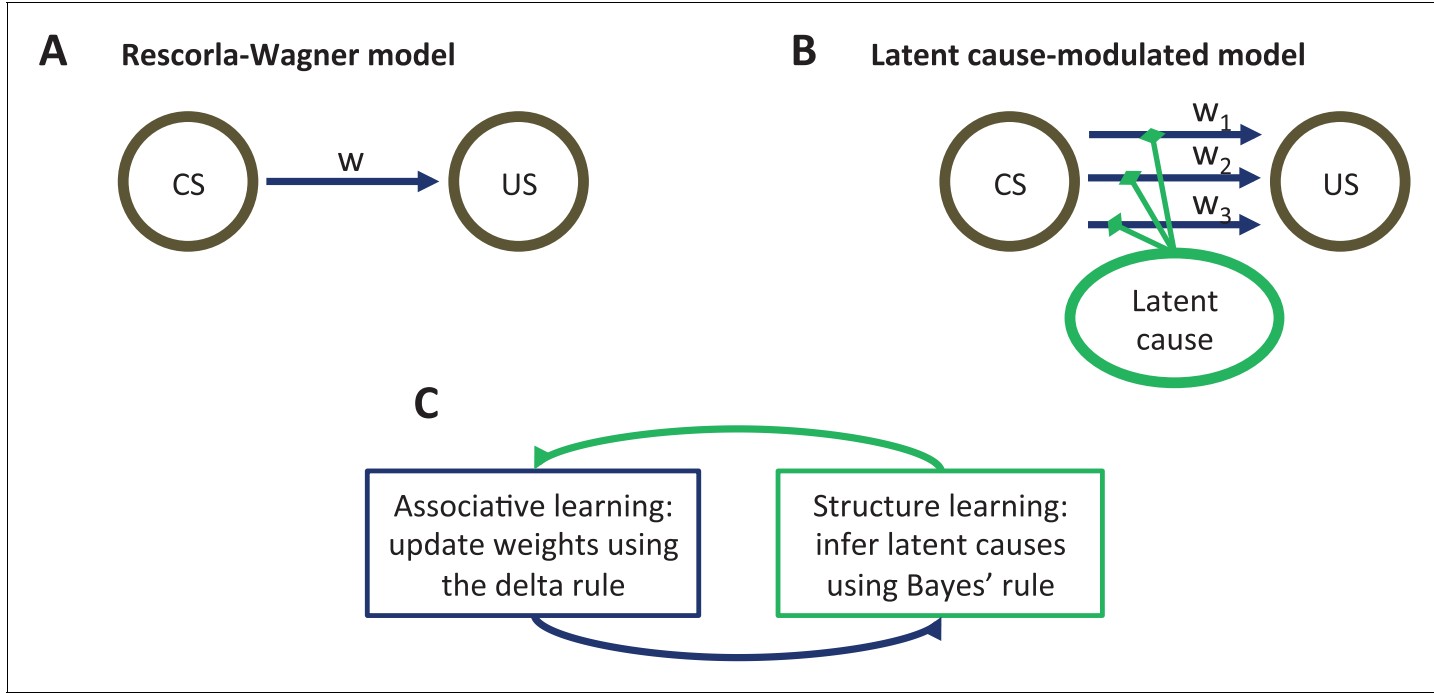

**Figure 1.** Model schematic. (**A**) The associative structure underlying the Rescorla-Wagner model. The associative strength between a conditioned stimulus (CS) and an unconditioned stimulus (US) is encoded by a scalar weight, $w$, that is updated through learning. (**B**) The associative structure underlying the latent-cause-modulated model. As in the Rescorla-Wagner model, associative strength is encoded by a scalar weight, but in this case there is a collection of such weights, each paired with a different latent cause. The US prediction is a linear combination of weights, modulated by the posterior probability that the corresponding latent cause is active. Alternatively, this model can be understood as consisting of three-way associations between the latent cause, the CS and the US. (**C**) A high-level schematic of the computations in the latent-cause model. Associative learning, in which the associative weights are updated (using the delta rule) conditional on the latent-cause posterior, alternates with structure learning, in which the posterior is updated (using Bayes' rule) conditional on the weights.

(*Figure 1B*). This allows the theory to move beyond the 'extinction=unlearning' assumption by positing that different latent causes are inferred during acquisition and extinction, and thus two different associations are learned (see also *Redish et al., 2007*). Memory modification arises when CS reexposure provides evidence to the animal that the latent cause assigned to the acquisition phase is once again active, making that cause's associations eligible for updating (or disruption by amnestic agents like PSIs).

The theory consists of two interacting sub-systems (*Figure 1C*): an associative-learning system updates a set of CS-US associations using a delta rule (*Rescorla and Wagner, 1972*; *Sutton and Barto, 1998*; *Widrow and Hoff, 1960*), while a structure learning system updates an approximation of the posterior distribution over latent causes using Bayes' rule. It is useful to envision the associative-learning system as almost identical to the Rescorla-Wagner model, with the key difference that the system can maintain multiple sets of associations between any CS-US pair (one for each latent cause; *Figure 1B*) instead of just a single set. Given a particular CS configuration (e.g., tone in a red box), the multiple associations are combined into a single prediction of the US by averaging the US prediction for each cause, weighted by the posterior probability of that cause being active. This posterior probability takes into account not only the conditional probability of the US given the CS configuration, but also the probability of observing the CS configuration itself. In the special case that only a single latent cause is inferred by the structure learning system, the associative learning system's computations are almost identical to the Rescorla-Wagner model (see the Materials and methods).

To infer the posterior distribution over latent causes, the structure learning system makes certain assumptions about the statistics of latent causes (the animal's 'internal model'). Informally, the main assumptions we impute to the animal are summarized by two principles:

- *Simplicity principle*: sensory inputs tend to be generated by a small (but possibly unbounded) number of latent causes. The simplicity principle, or Occam's razor, has appeared throughout cognitive science in many forms (*Chater and Vitányi, 2003*; *Gershman and Niv, 2013*). We use an 'infinite-capacity' prior over latent causes that, while preferring a small number of causes, allows the number of latent causes to grow as more data are observed (*Gershman and Blei, 2012*).
- *Contiguity principle*: the closer two events occur in time, the more likely it is that they were generated by the same latent cause. In other words, *latent causes tend to persist in time*. Spatial proximity also likely plays an important role in latent causal inference, but since this variable has not been thoroughly investigated in the memory modification literature, we omit it here for simplicity. See *Soto et al. (2014)* for an exploration of spatial proximity within a latent cause framework.

When combined with a number of auxiliary assumptions, these principles specify a complete generative distribution over stimulus configurations and latent causes—the animal's internal model. We now describe the theory in greater technical detail. In the section 'Understanding Extinction and Recovery,' we walk through a simple example with a single CS.

## The internal model

Our specification of the animal's internal model consists of three parts: (1) a distribution over latent causes, (2) a conditional distribution over CS configurations given latent causes, and (3) a conditional distribution over the US given the CS configuration. We now introduce these formally, starting from (2) and (3) and ending with (1). Let $\mathbf{x}_t = \{x_{t1}, \ldots, x_{tD}\}$ denote the $D$-dimensional CS configuration at time $t$, and let $r_t$ denote the US intensity at time $t$. In most of our simulations, we treat the US as binary (e.g., representing the occurrence or absence of a shock in Pavlovian fear conditioning). The distribution over $r_t$ and $\mathbf{x}_t$ is determined by a latent cause $z_t$. Specifically, the CS configuration is drawn from a Gaussian distribution:

$$P(\mathbf{x}_t | z_t = k) = \prod_{d=1}^{D} \mathcal{N}(x_{td}; \mu_{kd}, \sigma_x^2), \tag{1}$$

where $\mu_{kd}$ is the expected intensity of the $d$th CS given cause $k$ is active, and $\sigma_x^2$ is its variance. A Gaussian distribution was chosen for continuity with our recent modeling work (*Soto et al., 2014*; *Gershman et al., 2014*); most of our simulations will for simplicity use binary stimuli see [for a latent

cause theory based on a discrete stimulus representation] (*Gershman et al., 2010*). We assume a zero-mean prior on $\mu_{kd}$ with a variance of 1, and treat $\sigma_x^2$ as a fixed parameter (see the Materials and methods). Similarly to the Kalman filter model of conditioning (*Kakade and Dayan, 2002*; *Kruschke, 2008*), we assume that the US is generated by a weighted combination of the CS intensities corrupted by Gaussian noise, where the weights are determined by the latent cause:

$$P(r_t|z_t = k) = \mathcal{N}\left(r_t; \sum_{d=1}^{D} w_{kd}x_{td}, \sigma_r^2\right). \tag{2}$$

Finally, according to the animal's internal model, a single latent cause is responsible for all the events (CSs and USs) in any given trial. We will call this latent cause the *active* cause on that trial. A *priori*, which cause is the active latent cause on trial t, $z_t$, is assumed to be drawn from the following distribution:

$$P(z_t = k|\mathbf{z}_{1:t-1}) \propto \begin{cases} \sum_{t'<t} \mathcal{K}(\tau(t)-\tau(t'))\mathbb{I}[z_{t'}=k] & \text{if } k \leq K \text{ (i.e., } k \text{ is an old cause)} \\ \alpha & \text{otherwise (i.e., } k \text{ is a new cause)} \end{cases} \tag{3}$$

where $\mathbb{I}[\cdot] = 1$ when its argument is true (0 otherwise), $\tau(t)$ is the time at which trial $t$ occurred, $\mathcal{K}$ is a temporal kernel that governs the temporal dependence between latent causes, and $\alpha$ is a 'concentration' parameter that governs the probability of a completely new latent cause being responsible for the current trial. Intuitively, this distribution allows for an unlimited number of latent causes to have generated all observed data so far (at most $t$ different latent causes for the last $t$ trials), but at the same time, it is more likely that fewer causes were active. Importantly, due to the temporal kernel, the active latent cause on a particular trial is likely to be the same latent cause as was active on other trials that occurred nearby in time.

This infinite-capacity distribution over latent causes imposes the simplicity principle described in the previous section—a small number of latent causes, each active for a continuous period of time, is more likely *a priori* than a large number of intertwined causes. The distribution defined by *Equation 3* was first introduced by *Zhu et al. (2005)* in their 'time-sensitive' generalization of the Chinese restaurant process (*Aldous, 1985*). It is also equivalent to a special case of the 'distance dependent' Chinese restaurant process described by (*Blei and Frazier, 2011*). Variants of this distribution have been widely used in cognitive science to model probabilistic reasoning about combinatorial objects of unbounded cardinality (e.g., *Anderson, 1991*; *Sanborn et al., 2010*; *Collins and Frank, 2013*; *Goldwater et al., 2009*; *Gershman and Niv, 2010*). See *Gershman and Blei (2012)* for a tutorial introduction.

For the temporal kernel, we use a power law kernel:

$$\mathcal{K}(\tau(t)-\tau(t')) = \frac{1}{\tau(t)-\tau(t')}, \tag{4}$$

with $\mathcal{K}(0) = 0$. While other choices of temporal kernel are possible, our choice of a power law kernel was motivated by several considerations. First, it has been argued that forgetting functions across a variety of domains follow a power law (*Wixted and Ebbesen, 1991*; *Wixted, 2004*), and similar ideas have been applied to animal foraging (*Devenport et al., 1997*). While the temporal kernel is not literally a forgetting function, it implies that the CR strength elicited by a CS will decline as a function of the acquisition-test interval, because the probability that the acquisition and test trials were generated by different latent causes increases over the same interval. Thus, the temporal kernel induces a particular forgetting function that (all other things being equal) shares its shape.

Second, the power law kernel has an important temporal compression property, illustrated in *Figure 2*. Consider two timepoints, $t_1 < t_2$, separated by a fixed temporal interval, $\tau(t_2) - \tau(t_1)$, and a third time point, $t_3 > t_2$, separated from $t_2$ by a variable interval, $\tau(t_3) - \tau(t_2)$. In general, because $t_3$ is closer to $t_2$ than to $t_1$, the latent cause that generated $t_2$ is more likely to have also generated $t_3$, as compared to the latent cause that generated $t_1$ having generated $t_3$ (the contiguity principle). However, this advantage diminishes over time, and asymptotically disappears: as $t_1$ and $t_2$ both recede into the past relative to $t_3$, they become (almost) equally distant from $t_3$, and it is equally likely that one of their causes also caused $t_3$.

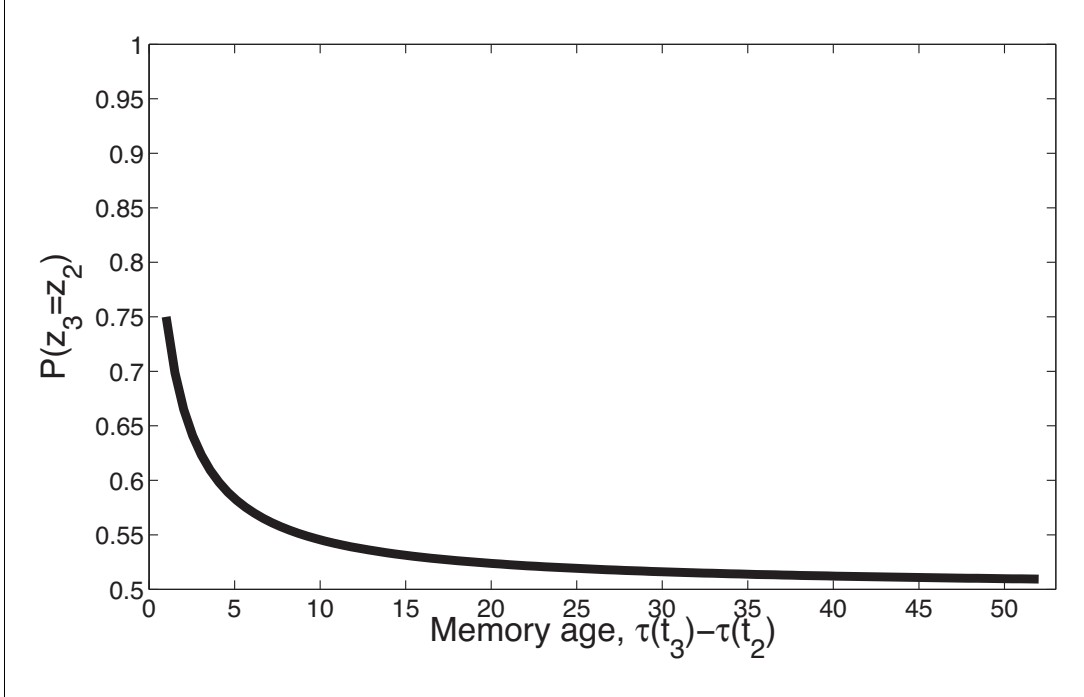

**Figure 2.** Temporal compression with the power law kernel. We assume that $t_1$ was generated by cause $z_1$, two timepoints later $t_2$ was generated by cause $z_2$, and a variable number of timepoints later $t_3$ was generated by cause $z_3$. To illustrate the time compression property we have assumed that the probability of a new cause is 0 (i.e., $\alpha = 0$) so inference at $t_3$ is constrained to one of the previous causes. As the temporal distance between $\tau(t_3)$ and the time of the previous trial $\tau(t_2)$ increases, that is, as the memory for $t_2$ recedes into the past, the probability of trial three being generated by either of the two prior latent causes becomes increasingly similar.

This completes our description of the animal's internal model. In the next section, we describe how an animal can use this internal model to reason about the latent causes of its sensory inputs and adjust the model parameters to improve its predictions.

## Associative and structure learning

In our framework, two computational problems confront the animal: (1) associative learning, that is, estimation of the model parameters (specifically, the associative weights, **W**) by maximizing the likelihood of the observed data given their hypothetical latent causes; and (2) structure learning, that is, determining which observation was generated by which latent cause, by computing the posterior probability for each possible assignment of observations to latent causes. One practical solutions is to alternate between these two learning processes. In this case, the learning process can be understood as a variant of the expectation-maximization (EM) algorithm (*Dempster et al., 1977*; *Neal and Hinton, 1998*), that has been suggested to provide a unifying framework for understanding cortical computation (*Friston, 2005*). We note at the outset that we do not necessarily think the brain is literally implementing these equations; more likely, the brain implements computations that have comparable functional properties. The question of neural mechanisms implementing these computations is taken up again in the Discussion. However, serial alternation of these two processes will be key to explaining the Monfils-Schiller findings.

For our model, the EM algorithm takes the following form (see Materials and methods for a derivation): after each observation, the model alternates between structure learning (the E-step, in which the posterior distribution over latent causes is updated assuming the current weights associated with the different causes are the true weights) and associative learning (the M-step, in which the weights for each cause are updated using a delta rule, conditional on the posterior over latent causes).

$$\text{E-step}: q_{tk}^{n+1} = P(z_t = k | \mathcal{D}_{1:t}, \mathbf{W}^n) \tag{5}$$

$$\text{M-step}: w_{kd}^{n+1} = w_{kd}^n + \eta x_{td} \delta_{tk}^{n+1} \tag{6}$$

for all latent causes $k$ and features $d$, where $n$ indexes EM iterations, $\eta$ is a learning rate and

$$\delta_{tk}^{n+1} = q_{tk}^{n+1}(r_t - \textstyle\sum_d w_{kd} x_{td}) \tag{7}$$

is the prediction error at time $t$ for latent cause $k$. The set of weight vectors for all latent causes at iteration $n$ is denoted by $\mathbf{W}^n$, and the CS-US history from trial 1 to $t$ is denoted by $\mathcal{D}_{1:t} = \{\mathbf{X}_{1:t}, \mathbf{r}_{1:t}\}$, where $\mathbf{X}_{1:t} = \{\mathbf{x}_1, \ldots, \mathbf{x}_t\}$ and $\mathbf{r}_{1:t} = \{r_1, \ldots, r_t\}$. Note that the updates are performed trial-by-trial in an incremental fashion, so earlier timepoints are not reconsidered.

Associative learning in our model (the M-step of the EM algorithm) is a generalization of the Rescorla-Wagner model (see the Materials and methods for further details). Whereas in the Rescorla-Wagner model there is a single association between a CS and the US (*Figure 1A*), in our generalization the animal forms multiple associations, one for each latent cause (*Figure 1B*). The overall US prediction is then a linear combination of the predictions of each latent cause, modulated by the posterior probability distribution over latent causes, represented by $q$ (see next section for details). Associative learning proceeds by adjusting the weights using gradient descent to minimize the prediction error.

Structure learning (the E-step of the EM algorithm) consists of computing the posterior probability distribution over latent causes using Bayes' rule:

$$P(z_t = k | \mathcal{D}_{1:t}, \mathbf{W}^n) = \frac{P(\mathcal{D}_{1:t} | z_t = k, \mathbf{W}^n) P(z_t = k)}{\sum_j P(\mathcal{D}_{1:t} | z_t = j, \mathbf{W}^n) P(z_t = j)}. \tag{8}$$

The first term in the numerator is the *likelihood*, encoding the probability of the animal's observations under the hypothetical assignment of the current observation to latent cause $k$, and the second term is the *prior* probability of this hypothetical assignment (*Equation 3*), encoding the animal's inductive bias about which latent causes are likely to be active. As explained in the Materials and methods, Bayes' rule is in this case computationally intractable (due to the implicit marginalization over the history of previous latent cause assignments, $\mathbf{z}_{1:t-1}$); we therefore use a simple and effective approximation (see *Equation 14*). In principle, the posterior computation requires perfect memory of all latent causes inferred in the past. Because temporally distal latent causes have vanishingly small probability under the prior, they can often be safely ignored, though solving this problem more generally may require a truly scale-invariant memory (see *Howard and Eichenbaum, 2013*).

Because the E and M steps are coupled, the learning agent needs to alternate between them (*Figure 1C*). We envision this process as corresponding to a kind of offline 'rumination,' in which the animal continues to revise its beliefs even after the stimulus has disappeared, somewhat similar to the 'rehearsal' process posited by *Wagner et al. (1973)*. In the context of Pavlovian conditioning, we assume that this rumination happens during intervals between trials, up to some maximum number of iterations (under the assumption that after a finite amount of time the animal will get distracted by something new and cease to ruminate on its past experience). In our simulations, we take this maximum number to be 3, where each iteration takes a single timestep. While the qualitative structure of the theory's predictions does not depend strongly on this maximum number, we found this to produce the best match with empirical data. The explanatory role of multiple iterations will play a key role in explaining the Monfils-Schiller findings.

## Conditioned responding

Given the learning model above, when faced with a configuration of CSs on trial $t$, the optimal prediction of the US is given by its expected value, averaging over the possible latent causes according to their posterior probability of currently being active:

$$\tilde{r}_t = \mathbb{E}[r_t | \mathbf{x}_t, \mathcal{D}_{1:t-1}] = \sum_{d=1}^{D} x_{td} \sum_k w_{kd} P(z_t = k | \mathbf{x}_t, \mathcal{D}_{1:t-1}, \mathbf{W}^n). \tag{9}$$

Most earlier Bayesian models of conditioning assumed that the animal's conditioned response is

directly proportional to the expected US (e.g., *Courville, 2006*; *Gershman and Niv, 2010*; *Kakade and Dayan, 2002*). In our simulations, we found that while *Equation 9* generally agrees with the direction of empirically observed behavior, the predicted magnitude of these effects was not always accurate. One possible reason for this is that in fear conditioning the mapping from predicted outcome to behavioral response may be nonlinear. Indeed, there is some evidence that freezing to a CS is a nonlinear function of shock intensity (*Baldi et al., 2004*). We therefore use a sigmoidal transformation of *Equation 9* to model the conditioned response:

$$\mathrm{CR} = 1 - \Phi(\theta; \tilde{r}_t, \lambda), \tag{10}$$

where $\Phi(\cdot; \tilde{r}_t, \lambda)$ is the Gaussian cumulative distribution function with mean $\tilde{r}_t$ and variance $\lambda$. One way to understand *Equation 10* is that the animal's conditioned response corresponds to its expectation that the US is greater than some threshold, $\theta$. When $\lambda = \sigma_r^2$ (the US variance), *Equation 10* corresponds precisely to the posterior probability that the US exceeds $\theta$:

$$\mathrm{CR} = P(r_t > \theta | \mathbf{x}_t, \mathcal{D}_{1:t}) = \int_{\theta}^{\infty} P(r_t | \mathbf{x}_t, \mathcal{D}_{1:t}) dr_t. \tag{11}$$

In practice, we found that more accurate results could be obtained by setting $\lambda < \sigma_r^2$. At a mechanistic level, $\lambda$ functions as an inverse gain control parameter: smaller values of $\lambda$ generate more sharply nonlinear responses (approaching a step function as $\lambda \to 0$). The parameter $\theta$ corresponds to the inflection point of the sigmoid.

## Modeling protein synthesis inhibition

Many of the experiments on post-retrieval memory modification used PSIs administered shortly after CS reexposure as an amnestic agent. We modeled PSI injections after trial $t$ by decrementing all weights according to: $\mathbf{w}_k \leftarrow \mathbf{w}_k(1 - q_{tk})$, that is, we decremented the weights for latent cause $k$ towards 0 in proportion to the posterior probability that cause $k$ was active on trial $t$. As we elaborate later, this is essentially a formalization of the *trace dominance principle* proposed by *Eisenberg et al. (2003)*: memories will be more affected by amnestic agents to the extent that they control behavior at the time of treatment.

It is important to note here that the physiological effect of PSIs is a matter of dispute (*Routtenberg and Rekart, 2005*; *Rudy et al., 2006*). For example, *Rudy et al. (2006)* have observed that anisomycin causes apoptosis; *Routtenberg and Rekart (2005)* describe numerous other effects of PSIs, including inhibition of negative regulators (which could actually *increase* protein synthesis), catecholamine function, and possibly neural activity itself. We restrict ourselves in this paper to exploring one possible pathway of action, but these other pathways merit further modeling.

## Understanding extinction and recovery

Before modeling specific experimental paradigms, in this section we lay out some general intuitions for how our model deals with extinction and recovery. In previous work (*Gershman et al., 2010*), we argued that the transition from acquisition to extinction involves a dramatic change in the statistics of the animal's sensory inputs, leading the animal to assign acquisition and extinction trials to different latent causes. The result of this partitioning is that the acquisition associations are not unlearned during extinction, and hence can be later recovered, as is observed experimentally (*Bouton, 2004*). Thus, according to our model, the key to persistent reduction in fear is to finesse the animal's sensory statistics such that the posterior distribution over latent causes favors assigning both acquisition and extinction phases to the same latent cause.

One way to understand the factors influencing the posterior distribution over latent causes is in terms of prediction error, the discrepancy between what the animal expects and what it observes. This term typically refers to a US prediction error (i.e., was the US predicted or not?), but our analysis applies to CS prediction errors as well: in our framework, anything that is not expected under the current latent cause evokes a prediction error.

The prediction error plays two roles in our model: it is an associative learning signal that teaches the animal how to adjust its associative weights, and it is a segmentation signal indicating when a new latent cause is active. When the animal has experienced several CS-US pairs during acquisition,

it develops an expectation that is then violated during extinction, producing a prediction error. Learning rules such as Rescorla-Wagner's are 'error-correcting' as they modify associations or values so as to reduce future prediction errors. In our model, however, the prediction error can be reduced in two different ways: either by associative learning (e.g., unlearning the CS-US association) or by structure learning (e.g., assigning the extinction trials to a new latent cause). Initially, the prior simplicity bias towards a small number of latent causes favors unlearning, but persistent accumulation of these prediction errors over the course of extinction eventually makes the posterior probability of a new cause high. Thus, our framework recapitulates and formalizes the idea that standard acquisition and extinction procedures eventually lead to the formation of two memories or associations, one for CS-US and one for CS-noUS.

The trade-off between the effects of prediction errors on associative and structure learning is illustrated in *Figure 3*. When prediction errors are small, the posterior probability of the acquisition latent cause is high (leading to modification of the original memory) but the amount of CS-US weight change is small as there is little discrepancy between what was predicted and what was observed; if prediction errors are very large, the posterior probability of the acquisition latent cause is low (leading to formation of a new memory), and the change in the weight corresponding to the original memory is again small. In theory, therefore, there should exist an intermediate 'sweet spot' for extinction learning where the prediction errors are large enough to induce considerable weight change but small enough to avoid inferring a new latent cause. Later we describe one behavioral paradigm (the Monfils-Schiller paradigm) that achieves this sweet spot (see also *Gershman et al., 2013*).

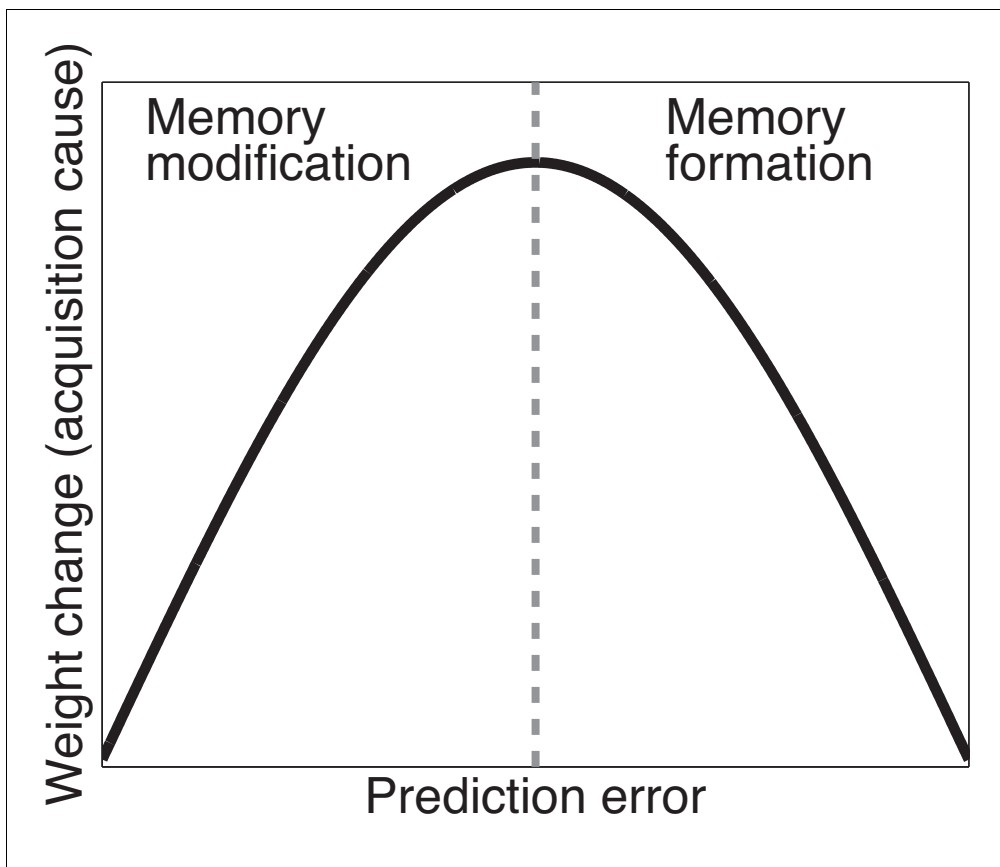

**Figure 3.** Cartoon of the model's predictions for fear extinction. The X-axis represents the size of the prediction error during extinction, and the Y-axis represents the change (after learning) in the weight for US prediction for the 'acquisition latent cause' (i.e., the latent cause inferred by the animal during conditioning).

To get a feeling for how the model's response to prediction errors depends on previous experience, consider a simple conditioning paradigm in which a single CS has been paired $N$ times with the US ($\mathcal{D}_{1:N} = \{x_t = 1, r_t = 1\}_{t=1}^{N}$), with a fixed ITI ($\tau(t) - \tau(t-1) = 1$). Under most parameter settings, this will result in all the acquisition trials being assigned to a single latent cause (hence we ignore the cause subscript $k$ in this example and refer to the single cause as the 'acquisition latent cause'). Now consider what happens when a single extinction trial ($x_{N+1} = 1, r_{N+1} = 0$) is presented. The posterior over latent causes (*Equation 5*) is proportional to the product of 3 terms: (1) The prior over latent causes, (2) the likelihood of the US, and (3) the likelihood of the CS. The third term plays a negligible role, since the CS does not change across acquisition and extinction, and hence no CS prediction error is generated. As $N$ grows, the prior probability of the acquisition latent cause generating the extinction trial as well increases, due to the simplicity bias of the prior over latent causes. However, associative learning of the weight vector counteracts this effect, since the US expectation, and hence the size of the prediction error due to the absence of the US (encoded in the likelihood term), also grows with $N$, asymptoting once the US prediction is fully learned. In particular, sensitivity to the US prediction error increases as $\sigma_r^2$ decreases (higher confidence in US predictions) and $\alpha$ increases (weaker simplicity bias). Parameter-dependence is examined systematically in the next section.

In order to understand some of the empirical phenomena described below, we must also explain why spontaneous recovery of the CR after extinction occurs. In our model, this occurs because the posterior probability of the acquisition cause increases as the extinction-test interval is lengthened, due to the temporal compression property of the power law temporal kernel $\mathcal{K}$ that we chose. As explained above, this kernel has the important property that older timepoints are 'compressed' together in memory: latent causes become more equiprobable under the prior as the time between acquisition and test increases. A similar idea was used by *Brown et al. (2007)* in their model of episodic memory to explain recency effects in human list learning experiments. Thus, the advantage of the extinction cause over the acquisition cause at test diminishes with the extinction-test interval. One implication of this analysis is that spontaneous recovery should never be complete, since the prior probability of the acquisition cause can never *exceed* the probability of the extinction cause (though the ratio of probabilities increases monotonically towards one as the extinction-test interval increases); this appears generally consistent with empirical data (*Rescorla, 2004*). There are a few examples of seemingly complete spontaneous recovery (*Quirk, 2002*; *Brooks and Bouton, 1993*; *Bouton and Brooks, 1993*). This is inconsistent with our theory and would require additional or different mechanisms, but it is currently unclear how common complete spontaneous recovery is, or what factors determine its completeness.

## Modeling post-retrieval memory modification

In this section, we show how our theory accounts for the basic data on post-retrieval memory modification. We seek to capture the *qualitative* pattern of results, rather than their precise quantitative form. We thus use the same parameter settings for all simulations (see Materials and methods), rather than fitting the parameters to data for each particular case. The parameter settings were chosen heuristically, but our results hold over a range of values.

*Schafe and LeDoux (2000)* showed that PSI administration immediately after the acquisition of a fear memory disrupted the CR in a later test phase; no disruption was found if the PSI administration was delayed. However, *Nader et al. (2000)* showed that re-exposing the animal to the CS prior to delayed PSI administration resulted in disrupted CR. The reexposure failed to produce this disruption if the PSI administration was delayed relative to the reexposure.

These observations are reproduced by simulations of our model (*Figure 4*). The key idea is that the latent cause inferred during the acquisition phase has high probability only after CS exposure (either during acquisition itself or after reexposure). Since we hypothesize that PSIs disrupt the associative weights tied to latent causes in proportion to their posterior probability (i.e., their degree of 'activation'), the model correctly predicts that PSI administration will be ineffective when delayed.

As discussed in the previous section, these effects are parameter-dependent. The key parameters of interest are the concentration parameter $\alpha$ and the US variance $\sigma^2$, which jointly determine sensitivity to prediction errors. In *Figure 5*, we show how variations in these parameters affect the CR in the Ret immediate condition of *Nader et al. (2000)*. As $\sigma^2$ increases, sensitivity to the post-retrieval

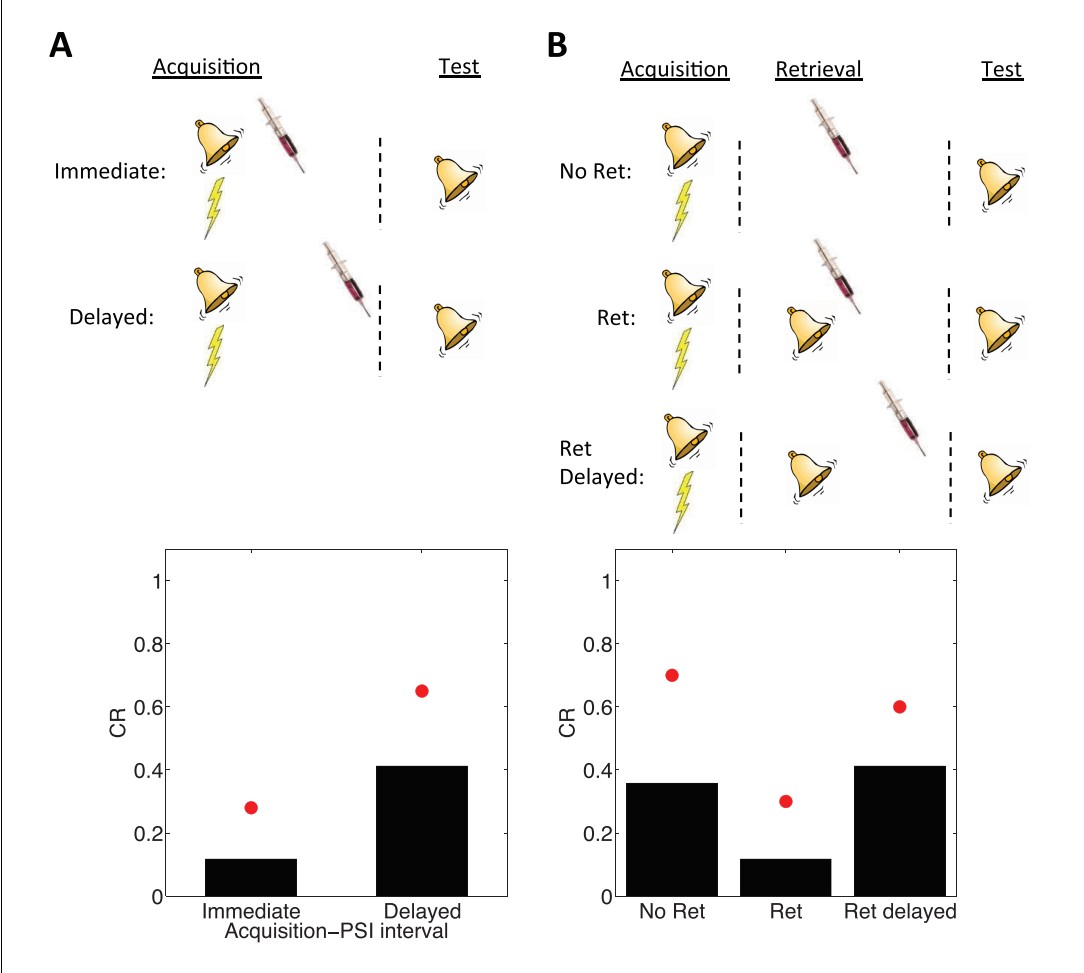

**Figure 4.** Simulation of post-retrieval memory modification. Top row shows a schematic of the experimental design (bell represents the tone CS, lightning bolt represents the shock US, syringe represents the injection of a protein synthesis inhibitor), with a conditioning → extinction → test structure. Bottom row shows model predictions in the test phase. (**A**) PSIs disrupt a fear memory (measured here through a freezing CR) when delivered immediately after the acquisition phase, but not when delivered after a delay. Red circles show proportion freezing of rats in the study by *Schafe and LeDoux (2000)*. (**B**) The delayed PSI administration is effective at disrupting the memory following reexposure to the CS (Ret). The effectiveness of this procedure is diminished if the PSI administration is delayed relative to reexposure (Ret delayed). Red circles show proportion freezing of rats in the study by *Nader et al. (2000)*.

prediction error decreases, thereby reducing the probability that a new latent cause will be inferred; this effect manifests as a reduced CR at test. The concentration parameter has a more complex effect on the results: increasing $\alpha$ initially increases the CR at test by increasing the probability that the post-retrieval trial is assigned to a new latent cause, but then decreases the CR at test due to the fact that—for sufficiently high values of $\alpha$—the test trial itself is assigned to an entirely new latent cause.

These simulations suggest empirically testable predictions. For example, our earlier work on context-dependent learning argued that young animals and animals with hippocampal lesions have small values of $\alpha$ (*Gershman et al., 2010*). Thus, these animals should show stronger retrieval effects. Other research has shown that individual differences in the propensity for generating latent causes (captured by fitting $\alpha$) can predict spontaneous recovery (*Gershman and Hartley, 2015*), suggesting that these individual differences should also be predictive of post-retrieval memory modification. Although less work has been done linking parametric differences in US variance to memory modification, recent work has argued that this parameter is encoded by striatal synapses expressing D1 and D2 receptors (*Mikhael and Bogacz, 2016*). Thus, we expect that pharmacological

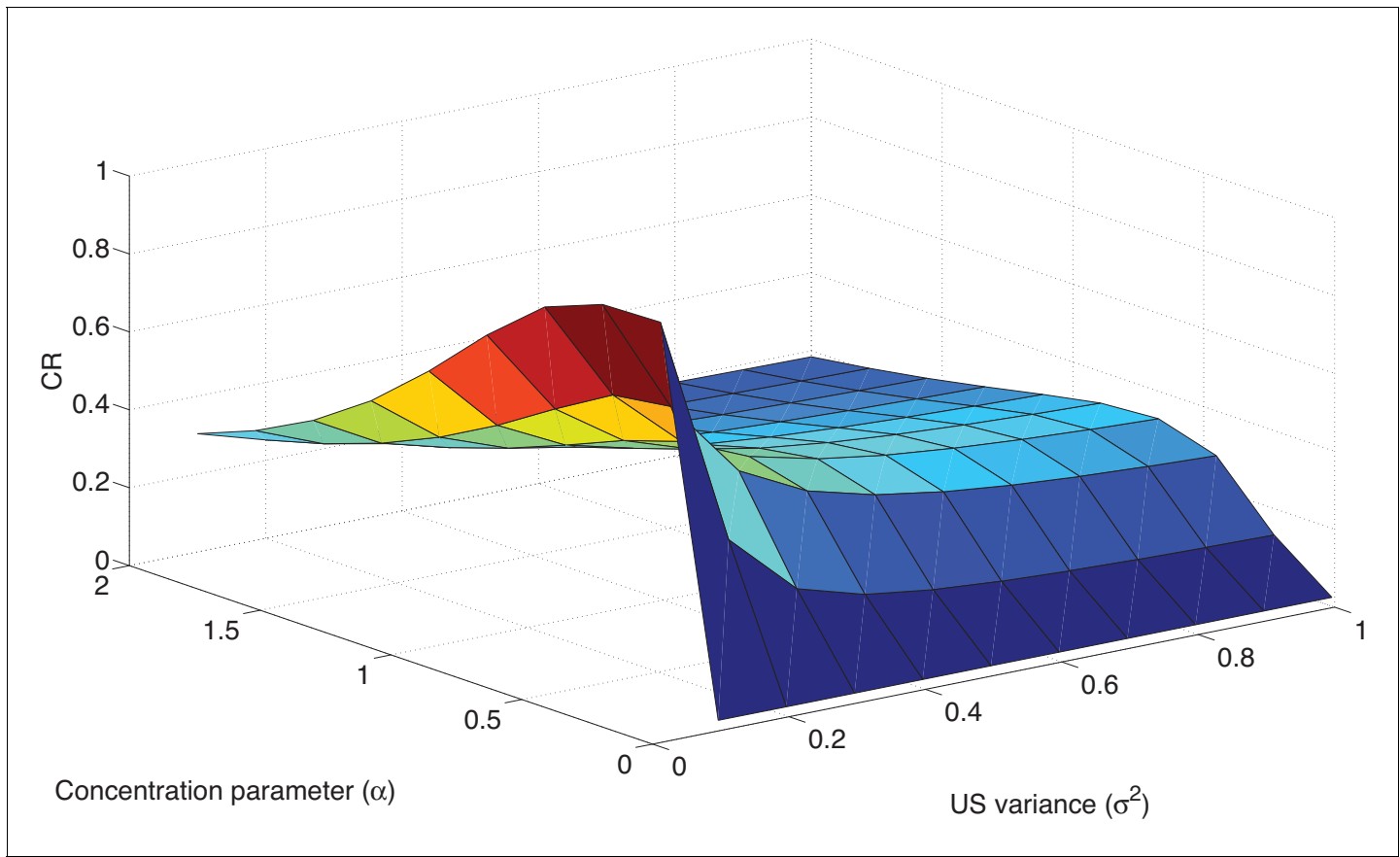

**Figure 5.** Parameter sensitivity in the Ret condition.

manipulations and individual variation of these receptors should be systematically related to post-retrieval memory modification.

We next explore several boundary conditions on post-retrieval memory modification see for a review (*Nader and Hardt, 2009*). Our goal is to show that these boundary conditions fall naturally out of our framework for Pavlovian conditioning.

## The 'trace dominance' principle

Using fear conditioning in the Medaka fish, *Eisenberg et al. (2003)* found that administering a PSI after a single reexposure to the CS (i.e., a single extinction trial) caused retrograde amnesia for the reactivated fear memory. In contrast, administering the PSI after multiple reexposures caused retrograde amnesia for the extinction memory: high recovery of fear was observed in a test on the following day. Similar results have been obtained with mice (*Suzuki et al., 2004*), rats (*Lee et al., 2006a*), and the crab *Chasmagnathus* (*Pedreira and Maldonado, 2003*). A 'trace dominance' principle interpretation of these data suggests that the extent of reexposure to the CS determines the dominance of a memory. In other words, the acquisition memory is initially dominant during reexposure (and hence vulnerable to disruption), but with repeated CS-alone exposure the extinction memory becomes dominant.

This is also seen in our theory: limited reexposure (operationalized by a single CS presentation) favors assignment of the reexposure trial to the acquisition latent cause. This follows from the simplicity bias in the latent cause prior: in the absence of strong evidence to the contrary, new observations are preferentially assigned to previously inferred causes. However, with more trials of extinction (e.g., two or more CS presentations), persistent prediction errors accrue, favoring assignment of these trials to a new latent cause (the 'extinction' cause). This logic leads to model predictions consistent with the empirical data (*Figure 6A*). Note that because ours is a trial-level model,

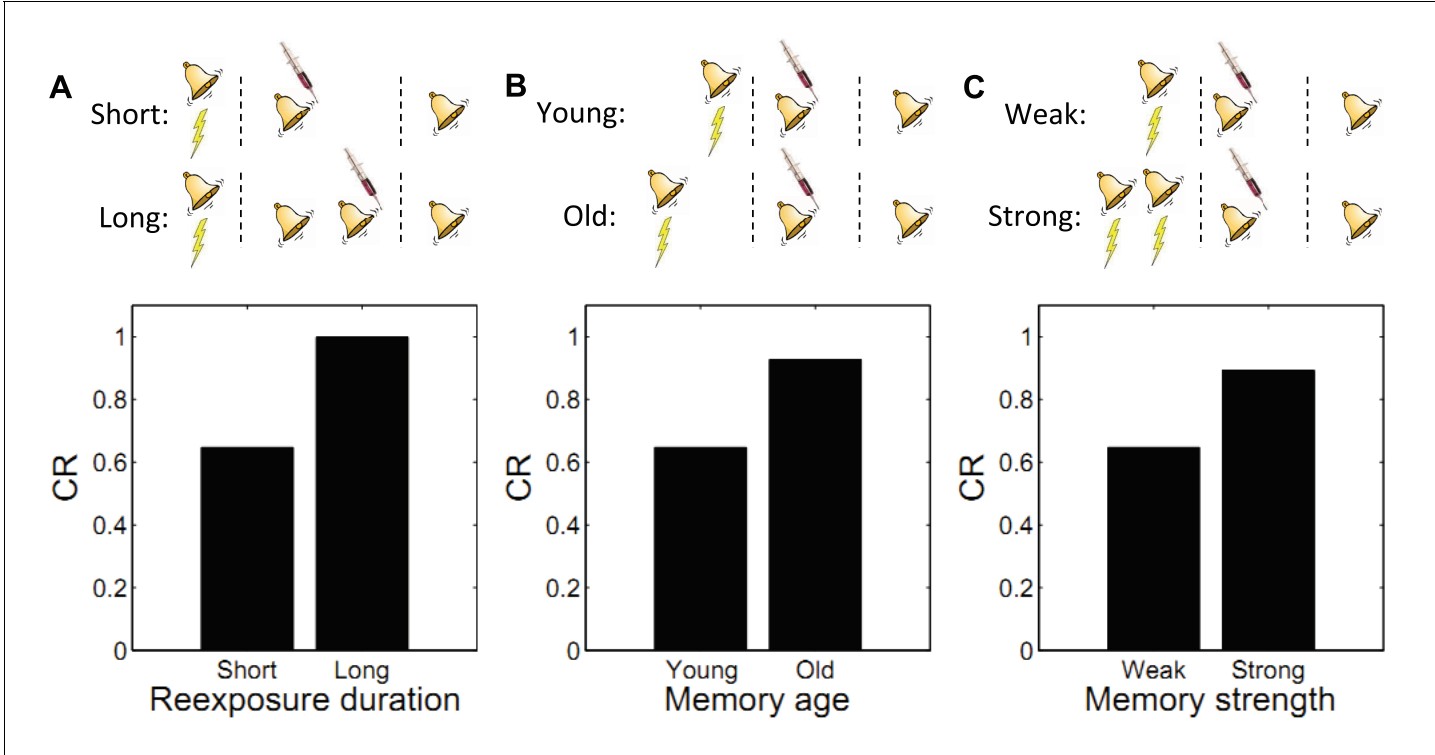

**Figure 6.** Boundary conditions on memory modification. Memory updating is attenuated under conditions of (**A**) more reexposure, (**B**) older or (**C**) stronger memories.

we cannot explicitly manipulate stimulus duration, so we use the number of presentations as a proxy.

## Memory age

By manipulating the interval between acquisition and reexposure, *Suzuki et al. (2004)* demonstrated that the amnestic effects of PSI injection were more pronounced for young memories (i.e., short intervals). *Winters et al. (2009)* found a similar effect with the NMDA receptor antagonist MK-801 administered prior to reexposure, and *Milekic and Alberini (2002)* demonstrated this effect in an inhibitory avoidance paradigm. *Alberini (2007)* has reviewed several other lines of evidence for the age-dependence of reconsolidation. These findings can also be explained by our model: old observations are less likely (under the prior) to have been generated by the same latent cause as recent observations. Thus, there is an inductive bias against modifying old memory traces. *Figure 6B* shows simulations of the Suzuki paradigm, demonstrating that our model can reproduce this pattern of results.

## Memory strength

In another experiment, *Suzuki et al. (2004)* showed that strong memories are more resistant to updating (see also *Wang et al., 2009*). Specifically, increasing the number of acquisition trials led to persistent fear even after reexposure to the CS and PSI injection. In terms of our model, this phenomenon reflects the fact that for stronger memories, because the associative weight is large, the prediction error is large, which causes the model to infer a new cause for the CS-alone trial. This new cause, in turn, would be the cause undergoing weakening due to PSI administration (i.e., the trace-dominance principle), rather than the old cause associated with the fear memory. Simulations of this experiment (*Figure 6C*) demonstrate that stronger memories are more resistant to updating in our model.

## Cue-specificity

*Doyère et al. (2007)* reported that disruption of memory by an amnestic treatment (in this case the mitogen-activated protein kinase inhibitor U0126) is restricted to a reactivated CS, leaving intact the CR to a non-reactivated CS that had also been paired with the US (*Figure 7A*). This finding is explained by observing that in our model learning only affects the CSs associated with the current inferred latent cause. In a recent study, *Debiec et al. (2013)* showed that cue-specificity of reconsolidation depends on separately training the two CSs; when they are trained in compound, reactivating one CS can render the other CS labile. Our model reproduces this effect (*Figure 7B*) as in this case the compound cue is assigned to a single latent cause that generates both CSs and the US, thereby coupling the two CSs.

In our simulation, responding at test is higher overall to the reactivated (relative to the non-reactivated) cue, in both the control and PSI conditions. The relatively higher responding to the reactivated cue is due to the fact that retrieval increases the probability of assigning the reactivated cue to the acquisition latent cause at test. This points to a discrepancy with the original data, since *Debiec et al. (2013)* did not find higher responding to the reactivated cue. Note, however, that this issue is orthogonal to the main point of interest in this study, namely the effect of PSI on reactivated and non-reactivated cues.

## Timing of multiple reexposures

When the same CS is reexposed twice with a relatively short (1 hr) ITI separating the presentations, PSI injection following the second presentation fails to disrupt the fear memory (*Jarome et al., 2012*). This is essentially another manifestation of the trace dominance principle (*Eisenberg et al., 2003*): two unreinforced reexposures cause the extinction trace to become more dominant, and the PSI therefore disrupts the extinction trace rather than the fear trace. *Jarome et al. (2012)* found that increasing the ITI between retrievals (from 1 hr to 6 hr, 24 hr and 1 week) resulted in a

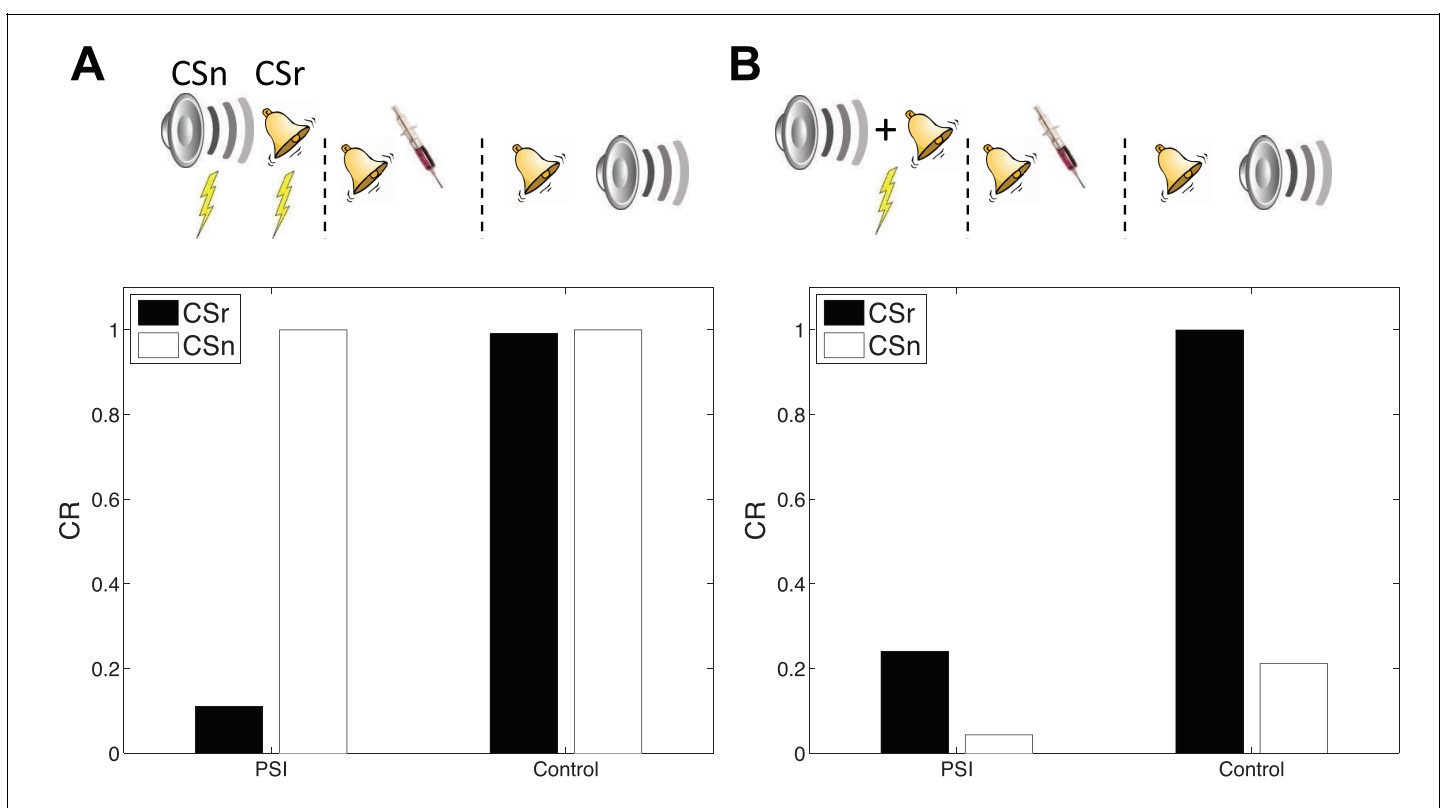

**Figure 7.** Cue-specificity of amnestic treatment. (**A**) Disruption of memory modification by amnestic treatment affects the reactivated cue (CSr) but not the non-reactivated cue (CSn). (**B**) When trained in compound, reactivating CSr renders CSn vulnerable to disruption of modification.

parametric decrease of fear at test, suggesting that longer intervals lead to disruption of the fear trace by the PSI. This effect is predicted by our theory, due to the time-dependent prior over latent causes, which prefers assigning trials separated by a long temporal interval to different causes. As a result, longer ITIs reduce the probability that the two reexposures were generated by the same 'extinction' latent cause, concomitantly increasing the probability that the second reexposure was generated by the 'acquisition' latent cause as compared to yet another new latent cause (*Figure 8*). This result is parameter-dependent: If the concentration parameter is sufficiently large, then increasing the interval will cause the animal to infer a new latent cause (different from the acquisition and extinction causes) and thus the acquisition cause will not be affected by the PSI. Note that in the *Jarome et al. (2012)* study, the retrieval-test interval, but not the acquisition-test interval, was fixed; thus their results may partly reflect time-dependent changes in the posterior over latent causes, as reflected in the control simulation.

## Transience of amnesia

A major focus of theories of experimental amnesia (i.e., forgetting of the association formed during acquisition) has been the observation that, under a variety of circumstances, recovery from amnesia can occur (*Miller and Matzel, 2006*; *Riccio et al., 2006*). A study by *Power et al. (2006)* provides a clear demonstration: Following conditioning, post-retrieval intrahippocampal infusions of the PSI anisomycin reduced conditioned responding when the rats were tested 1 day later, but responding recovered when the test was administered after sic days. Thus, the PSI-induced memory impairment was transient (see also *Lattal and Abel, 2004*). As pointed out by *Gold and King (1974)*, recovery from amnesia does not necessarily mean that the amnesia was purely a retrieval deficit. If the amnestic agent diminished, but did not entirely eliminate, the reactivated memory, then subsequent

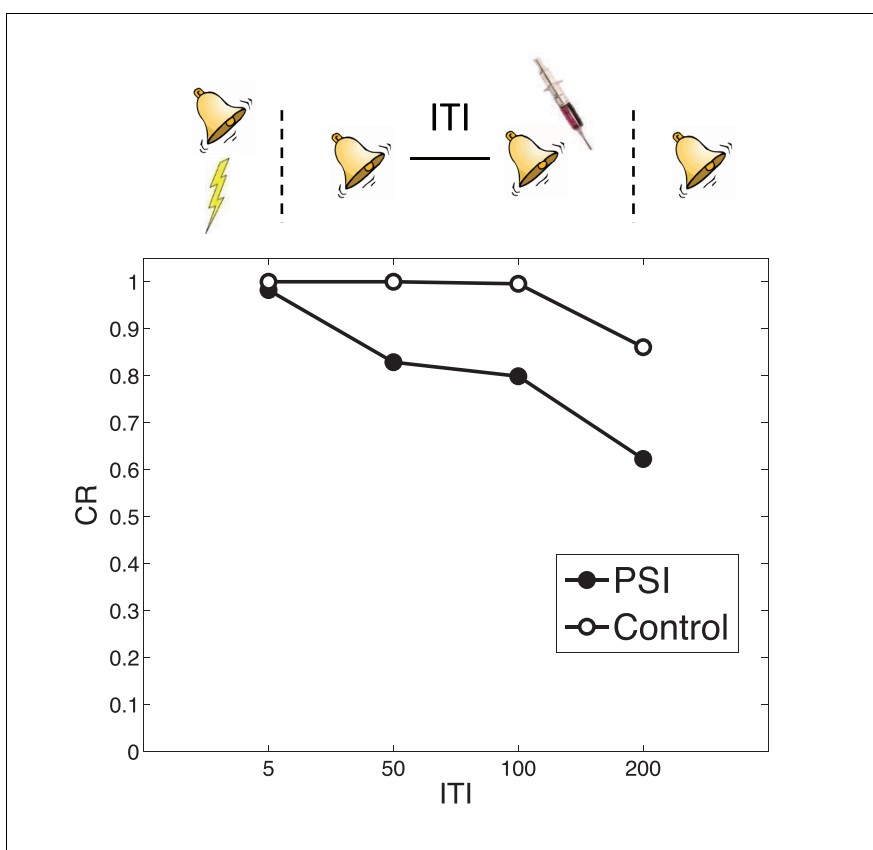

**Figure 8.** Timing of multiple reexposures. Lengthening the intertrial interval (ITI) between multiple reexposures increases the simulated effectiveness of PSI administration in attenuating fear at test. The control simulation shows results without PSI administration.

recovery could reflect new learning added on to the residual memory trace (under the assumption that memory reactivation itself supplies a learning experience).

The explanation that our theory offers for the transience of amnesia is related to Gold's interpretation, in that we also assume a residual memory trace. Since the amnestic agent does not entirely eliminate the memory trace, later recovery of the fear memory occurs because the relative probability of assigning a new test observation to the acquisition cause rather than to the cause associated with the retrieval session (which was, in effect, a short extinction session) increases over time (a consequence of temporal compression by the power law kernel, as explained above). In other words, this is a form of spontaneous recovery: The original (weakened) memory becomes more retrievable over time. Thus, our explanation can be viewed as a retrieval-based theory (see *Miller and Springer, 1974*), while not ruling out the possibility that the memory is partially degraded by the amnestic agent. Simulations shown in *Figure 9* demonstrate that this explanation can account for the increase in CR with longer retrieval-test intervals. Interestingly, the model predicts that further increasing the retrieval-test interval will eventually result in slightly reduced responding, because of the increased probability of a new latent cause at test.

## State-dependency

A long-standing explanation for recovery from amnesia is state-dependency: the idea that the internal state induced by the amnestic agent becomes part of the memory representation, such that disrupted responding at test can be explained by retrieval failure or generalization decrement (*Miller and Springer, 1973*; *Riccio et al., 2006*; *Spear, 1973*) due to the absence of the amnestic agent at test. In other words, apparent 'amnesia' is a consequence of a mismatch between internal states at acquisition and test. As pointed out by *Nader and Hardt (2009)*, this hypothesis faces a number of difficulties in explaining the empirical data. For example, it cannot explain why memory

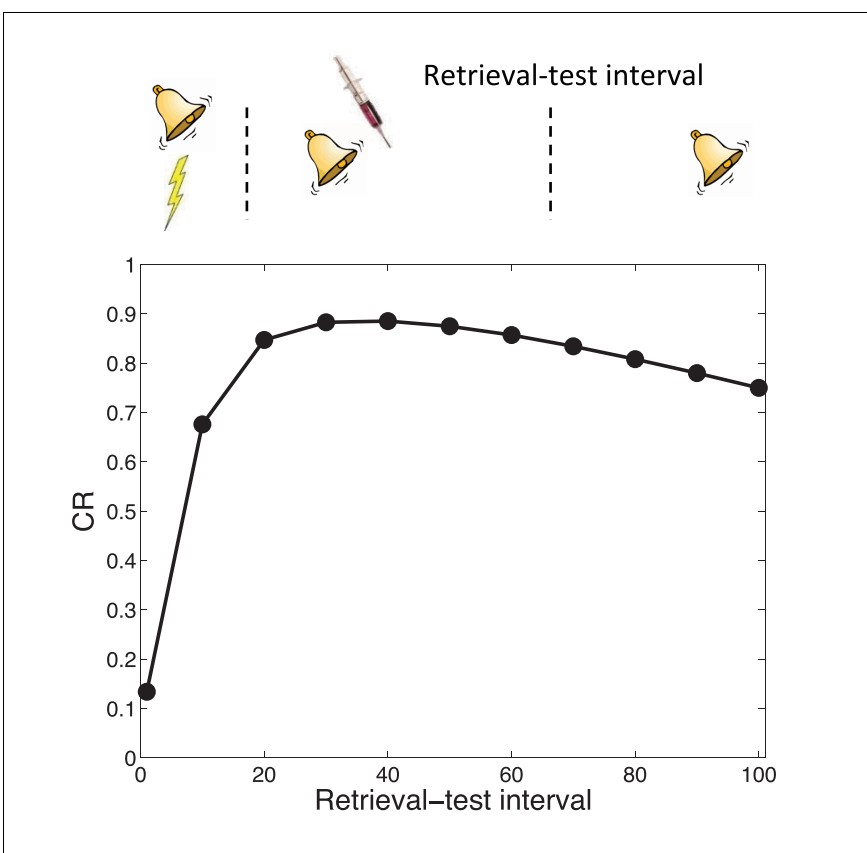

**Figure 9.** Transience of amnesia. Lengthening the interval between retrieval and test results in recovery from amnesia.

can sometimes be *enhanced* by post-retrieval treatments (e.g., *Tronson et al., 2006*; *Lee et al., 2009*; *Lee et al., 2006b*).

In spite of these difficulties, a recent report (*Gisquet-Verrier et al., 2015*) found striking evidence in favor of the state-dependency hypothesis (see also *Hinderliter et al., 1975*). As in previous studies, the authors found that post-conditioning PSI administration disrupted the CR at test; the novel twist was that administering the PSI both immediately after conditioning and immediately before test eliminated the disruptive effect. This finding fits naturally with the idea that the PSI induced a discriminative internal state, and hence the observed 'amnesia' was in fact a retrieval failure or generalization decrement.

We simulated state-dependency by adding the PSI as an additional CS (*Figure 10*). Consistent with the experimental findings of *Gisquet-Verrier et al. (2015)*, the state-dependent version of the latent cause theory reproduced the reversal of CR disruption by PSI administration prior to the test phase. Our other results are qualitatively unchanged when we add this additional state feature. Importantly, the state-dependency does not depend on any weakening effects of the PSI itself. Thus, a fairly simple extension of our model can accommodate the state-dependency hypothesis.

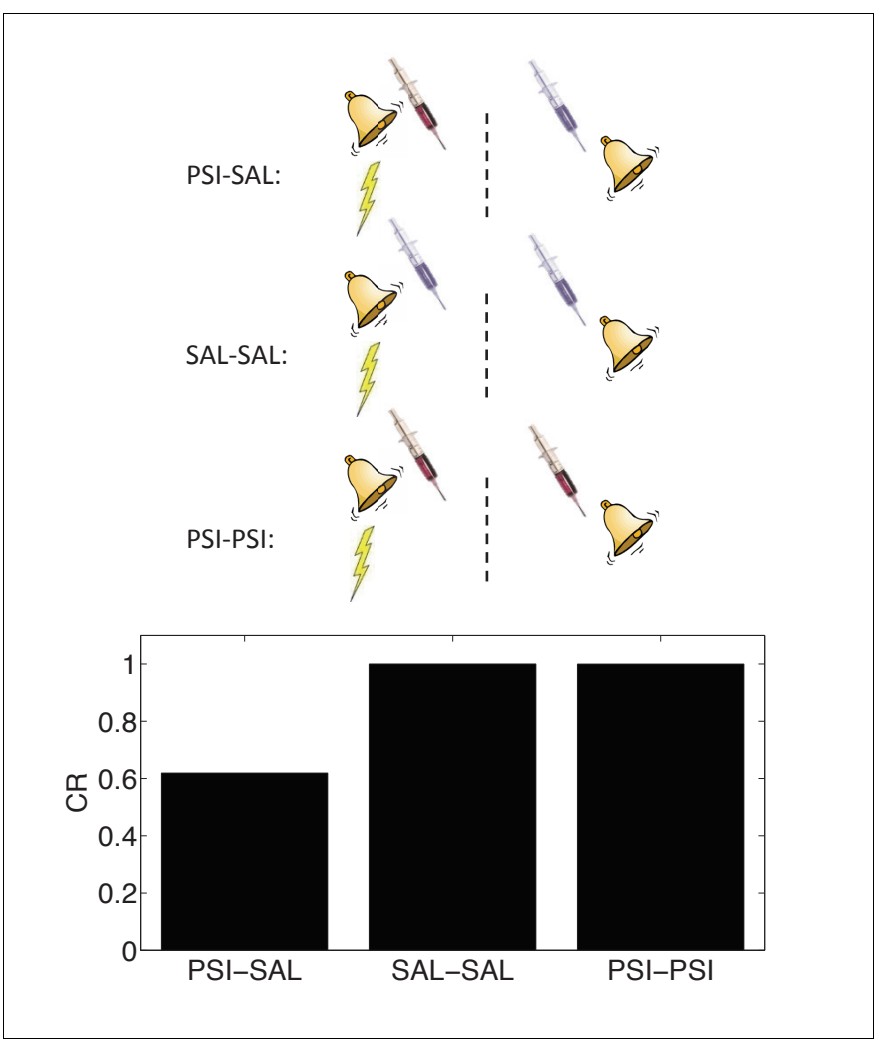

**Figure 10.** State-dependency of amnesia. The amnestic affect of PSI administration after conditioning can be reversed by readministering the PSI at the time of test ('PSI-PSI'). Here 'SAL' denotes administration of saline instead of the PSI, indicated by the pale syringe in the schematic.

## The Monfils-Schiller paradigm

While extinction procedures after fear conditioning are, in general, not effective in producing permanent and generalizable reduction of fear, two influential studies (*Monfils et al., 2009*; *Schiller et al., 2010*) demonstrated that a single reexposure ('retrieval trial') of a CS that had been associated with a shock, 24 hr after acquisition and 10–60 min before extinction, leads to persistent reduction of fear as measured by renewal, reinstatement and spontaneous recovery tests. Importantly, this effect did not require pharmacological interventions such as PSIs, and it was evident in both rodents (*Monfils et al., 2009*) and humans (*Schiller et al., 2010*).

These studies also revealed that: (1) reduction of fear in humans is still evident a year later; (2) the reduction is specific to the cue-reactivated memory; and (3) increasing the retrieval-extinction interval to 6 hr eliminates the effect. That is, extinction after a retrieval trial is more effective at modifying the original association than regular extinction, but this only holds for extinction sessions administered relatively promptly after the retrieval cue. This latter finding suggests that the retrieval cue engages a time-limited plasticity window, in which extinction operates. These findings have been replicated several times in rodents (*Auchter et al., 2017*; *Baker et al., 2013*; *Clem and Huganir, 2010*; *Jones et al., 2013*; *Olshavsky et al., 2013b*, *2013a*; *Ponnusamy et al., 2016*; *Rao-Ruiz et al., 2011*) and humans (*Agren et al., 2012*; *Oyarzún et al., 2012*; *Schiller et al., 2013*; *Steinfurth et al., 2014*), though the generality of the paradigm remains controversial (*Chan et al., 2010*; *Costanzi et al., 2011*; *Kindt and Soeter, 2013*; *Soeter and Kindt, 2011*; *Kredlow et al., 2016*).

It is important to recognize that the so-called 'retrieval trial' is operationally no different from an extinction trial—it is a CS presented alone. Essentially, the principal salient difference between the Monfils-Schiller paradigm and regular extinction training is that in the Monfils-Schiller paradigm, the interval between the first and second extinction trials is substantially longer than the intervals between all the other trials. Another difference (which we address later) is that in the Monfils-Schiller paradigm, the subject spends the retrieval-extinction interval outside the acquisition context, in its home cage. This phenomenon is thus puzzling for most—if not all—theories of associative learning. What happens during this one interval that dramatically alters later fear memory? Below we provide a normative computational account of this phenomenon based on our framework for Pavlovian conditioning. We also suggest explanations for some of the inconsistencies across studies.

Model simulations of the Monfils-Schiller paradigm are shown in *Figure 11*. We simulated three conditions, differing only in the retrieval-extinction interval (REI): *No Ret* (REI = 0, that is, extinction begins with no separate retrieval trial), *Ret-short* (REI = 3), *Ret-long* (REI = 100). Time is measured in arbitrary units here; see the Materials and methods for a description of how these units roughly map on to real time. As observed experimentally, in our simulations all groups ceased responding by the end of extinction. Both Ret-long and No Ret showed spontaneous recovery after a long extinction-test delay. In contrast, in the Ret-short condition there was no spontaneous recovery of fear at test. Examining the posterior distributions over latent causes in the different conditions (*Figure 11B–D*), we see that the extinction trials were assigned to a new latent cause in the No Ret and Ret-long conditions, but to the acquisition cause in the Ret-short condition.

Our theoretical explanation of data from the Monfils-Schiller paradigm rests critically on the 'rumination' process (i.e., iterative updating according to the EM algorithm) that occurs in the interval between the first and second extinction trials (the REI). Since there is some probability that the original ('acquisition') latent cause is active during the REI, the first iteration of associative learning in the EM algorithm will reduce the CS-US association for that latent cause. On the next iteration, the model will be even more likely to infer that the original latent cause is active (since the CS-US association strength is smaller, the prediction error induced by the CS appearing without the US will be even smaller). As a result of this increased probability that the original latent cause is active, the CS-US association will be reduced even more. In our model, the number of EM iterations (up to a maximum of 3 iterations, with one iteration per timestep; see model description) depends on the length of the REI. More iterations cause the original association to be further weakened after the first retrieval trial, and therefore spontaneous recovery of the original fear memory at test is attenuated (*Figure 12A*).

When the interval is too short (as in the No Ret condition), there is insufficient time (i.e., only a single EM iteration) to reduce the CS-US association and thus later extinction trials are preferentially

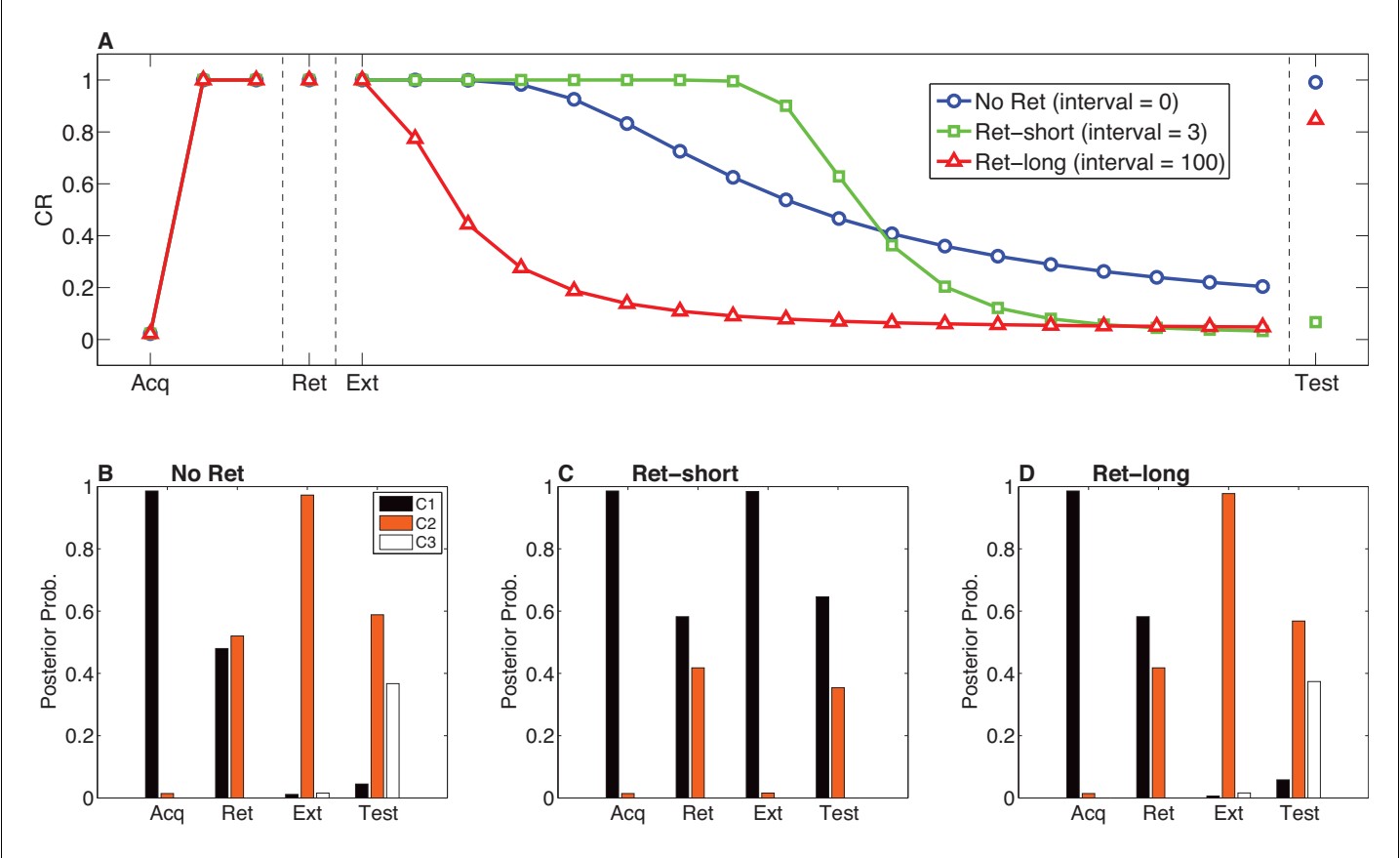

**Figure 11.** Model predictions for the Monfils-Schiller paradigm. (**A**) Simulated conditioned response (CR) during acquisition (Acq; 3 CS-US pairs), retrieval (Ret; 1 CS presentation 24 hr after acquisition, followed by no interval, a short interval, or a long interval before the next phase), extinction (Ext; CS-alone presentations) and a test phase 24 hr later. Three conditions are shown: No-Ret (no interval between retrieval and extinction; the 'Ret' trial depicted here is the first trial of extinction), Ret-short (retrieval with a short post-retrieval interval), and Ret-long (retrieval with a long post-retrieval interval). (**B–D**) The posterior probability distribution over latent causes (denoted C1, C2 and C3) in each condition. Probabilities for only the top three highest-probability causes are shown.

assigned to a new latent cause rather than the acquisition cause. When the interval is too long (as in the Ret-long condition), although the CS-US association of the acquisition latent cause is reduced during the REI, extinction trials will be preferentially assigned to a new latent cause due to the time-sensitive prior that suggests that events far away in time are generated by different causes. Thus in this condition as well, the original association is not attenuated by the extinction trials any further, and spontaneous recovery of fear occurs at test. It is only in the intermediate condition, the short REI, that the EM iterations reduce the prediction of the US by the acquisition latent cause sufficiently so as to allow later extinction trials to be assigned to this same latent cause, therefore reducing the prediction of the US by this cause even further, effectively 'erasing' the fear memory. This nonmonotonic dependence on the REI is shown in *Figure 12B*.

Note that our model predicts that all the boundary conditions discussed earlier should apply to the Monfils-Schiller paradigm. Thus, for example, older memories should be more difficult to disrupt, even with an intermediate REI. We revisit this point in the next section.

The importance of iterative adjustment during the retrieval-extinction interval suggests that distracting or occupying animals during the interval should disrupt the Monfils-Schiller effect. For example, our theory predicts that giving rodents a secondary task to perform during the interval will prevent the iterative weakening of the CS-US association of the acquisition cause, leading to assignment of extinction trials to a new latent cause (as in regular extinction) and to later recovery of fear. Alternatively, it might be possible to enhance the effect by leaving the animal in the conditioning

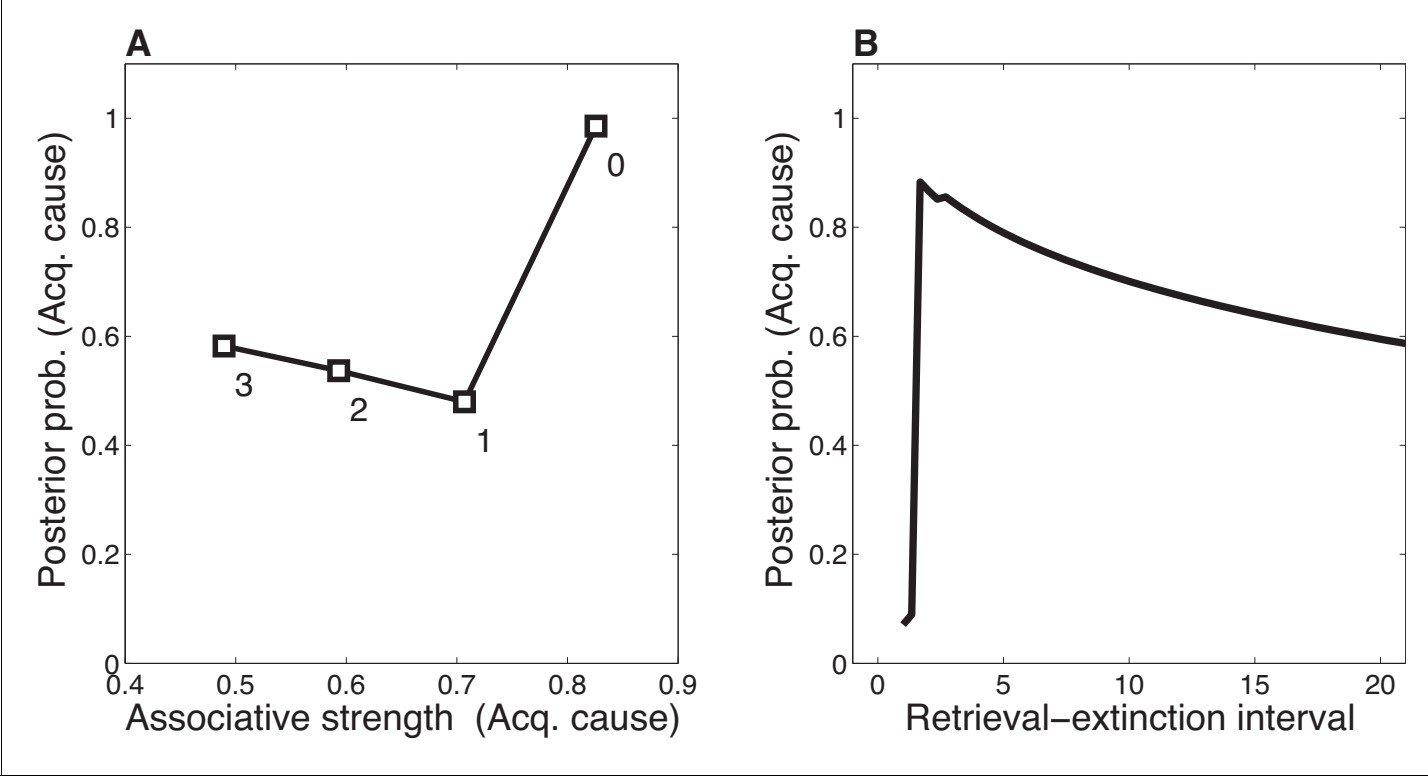

**Figure 12.** Dynamics of associative and structure learning during the retrieval-extinction interval in the Monfils-Schiller paradigm. (**A**) The X-axis represents the associative weight corresponding to the acquisition latent cause. The Y-axis represents the posterior probability that the acquisition latent cause is active for the retrieval trial. Each numbered square indicates a particular iteration during the retrieval-extinction interval, with '0' indicating the last trial of acquisition. Initially, the prediction error causes the posterior to favor a new latent cause rather than the old acquisition cause, however, over the course of three iterations, incremental reductions in the associative weight pull the posterior probability higher by making the retrieval trial more likely under the acquisition cause. (**B**) As the retrieval-extinction interval grows longer, the probability of assigning the first extinction trial to the acquisition cause changes non-monotonically. Two non-reinforced trials very close in time are likely to come from a new latent cause, thus the posterior probability of the acquisition cause generating these trials starts low. It peaks at a larger retrieval-extinction interval; as this interval increases, the acquisition cause's associative strength is incrementally reduced, thereby making the extinction trials more likely under the acquisition cause. The curve then gradually diminishes due to the time-sensitive prior that causes temporally separated events to be more likely to be generated by different causes (*Equation 3*). Each EM iteration takes a single timestep, and at least 1 EM iteration is always performed, up to a maximum of 3, depending on the intertrial interval.

chamber during the interval; the chamber would serve as a reminder cue, potentially preventing the animal from getting distracted. On the other hand, the conditioning chamber is associated with stress, so rumination may be about shocks rather than their absence. Thus, it might be that rumination in the safe haven of the homecage is most effective at producing subsequent memory modification.

## Cue-specificity in the Monfils-Schiller paradigm

*Figure 13* shows simulations of the cue-specificity experiment reported in *Schiller et al. (2010)*. In a within-subjects design, two CSs were paired with shock, but only one was reexposed prior to extinction in a 'retrieval' trial. Consistent with the findings of *Doyère et al. (2007)*, *Schiller et al. (2010)* found that fear recovered for the CS that was not reexposed, but not for the reexposed CS. This finding fits with our theoretical interpretation that CS reexposure leads to memory modification for the US association specific to that CS and the reactivated latent cause.

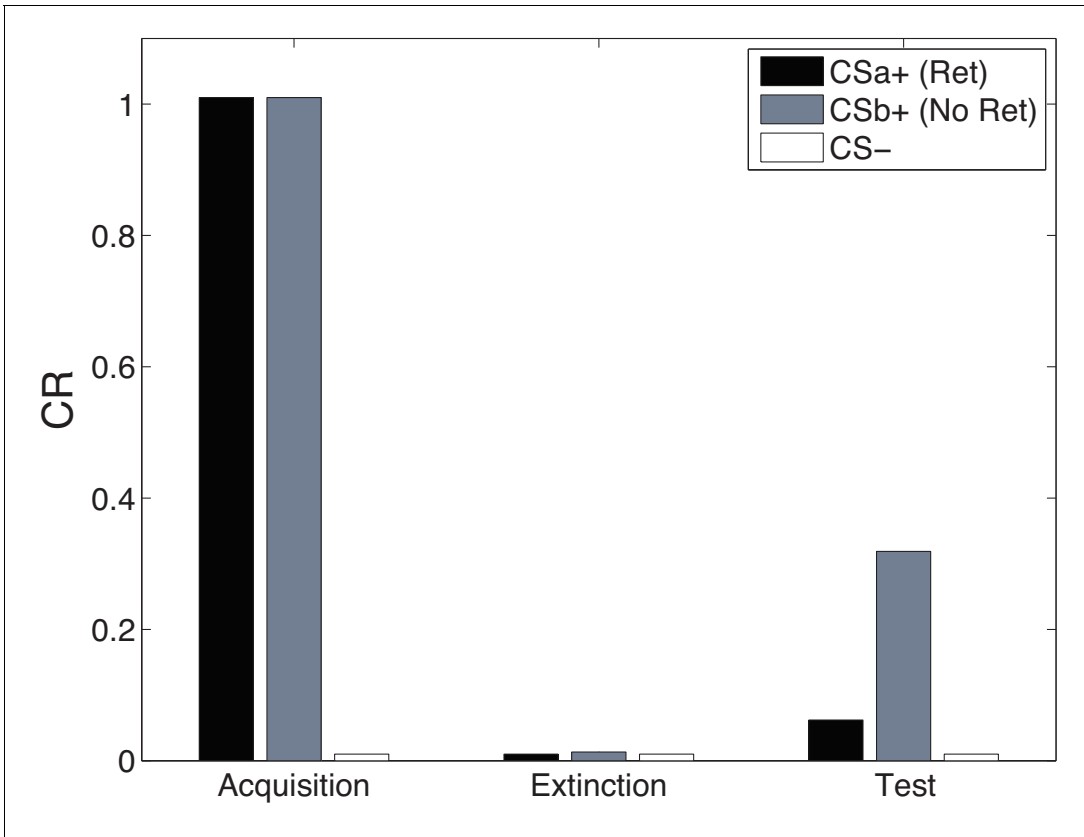

**Figure 13.** Cue-specificity in the Monfils-Schiller paradigm. Model simulations of the within-subjects design reported by *Schiller et al. (2010)*, in which two CSs (CSa+ and CSb+) were individually paired with shock (CS− was never paired with a shock), but only one (CSa+) was reexposed in a 'retrieval' trial prior to extinction. Fear recovery is attenuated for the reexposed CS.

### Boundary conditions in the Monfils-Schiller paradigm

Several studies have failed to show persistent reduction of fear using the Monfils-Schiller paradigm (*Chan et al., 2010*; *Costanzi et al., 2011*; *Kindt and Soeter, 2013*; *Soeter and Kindt, 2011*). Is the paradigm inherently fragile, or do these discrepancies delineate systematic boundary conditions? *Auber et al. (2013)* identified many methodological differences between experiments using the Monfils-Schiller paradigm. The question facing our theory is whether the effects of these differences can be explained as a rational consequence of inference given sensory data. Many of the differences involve experimental variables that are outside the scope of our theory, such as the tone frequency in auditory fear conditioning (*Chan et al., 2010*), or affective properties of picture stimuli in human studies (*Kindt and Soeter, 2013*; *Soeter and Kindt, 2011*), neither of which are explicitly represented in our model. We therefore focus on the two methodological differences that do fall within the scope of our theory.

*Costanzi et al. (2011)* trained mice to associate a foot-shock with a context, and then induced retrieval of the contextual memory 29 days later by placing the mice in the conditioning context for 3 min. An extinction session in the same context followed one hour later. The next day, the mice were tested for contextual fear in the conditioning context. *Costanzi et al. (2011)* found that extinction after retrieval did not attenuate contextual fear any more so than regular extinction without a retrieval trial, contrary to the findings of *Monfils et al. (2009)*.

*Auber et al. (2013)* pointed out that a crucial difference between the studies of *Costanzi et al. (2011)* and *Monfils et al. (2009)* was the acquisition-retrieval interval: 29 days in *Costanzi et al. (2011)* and 1 day in *Monfils et al. (2009)*. As we reviewed above, it is well-established that older memories resist modification (*Alberini, 2007*). According to our theory, this phenomenon occurs

because when the acquisition-retrieval interval is long, the retrieval trial is less likely to have been generated by the same latent cause as the acquisition trials.

Our theory therefore predicts the difference between the two experiments (*Figure 14A*). A direct comparison of new and old memories in the Monfils-Schiller paradigm was reported by (*Gräff et al., 2014*); consistent with our hypothesis, younger memories were more susceptible to modification (see also *Jones and Monfils, 2016*, for converging evidence).

In another study reporting discrepant results, *Chan et al. (2010)* found that the Monfils-Schiller paradigm failed to prevent the return of fear in renewal and reinstatement tests. *Auber et al. (2013)* observed that the study by *Chan et al. (2010)* used different experimental boxes located in different rooms for acquisition and for retrieval and extinction, whereas in their renewal experiment *Monfils et al. (2009)* modified the conditioning box to create a new context for retrieval and extinction, while keeping the room the same. We simulated the different contexts by adding a 'context' feature that allowed us to parametrically vary the similarity between acquisition and retrieval/extinction contexts. In particular, we assumed that this feature was 1 in acquisition, and then in retrieval and extinction we represented the similar context by setting the feature to 0.8, whereas the dissimilar context feature was set to 0. We found that retrieval and extinction in a similar context led to less renewal at test than did retrieval and extinction in a very different context (*Figure 14B*). This is because when acquisition and retrieval/extinction contexts are similar, there is a higher probability that the latter trials will be assigned to the original acquisition latent cause—i.e., that the 'retrieval' trial will indeed retrieve the old association (see also *Gershman et al., 2010*, *2013c*). As with the *Costanzi et al. (2011)* study, it is important to note that (*Chan et al., 2010*) did not parametrically manipulate similarity, so further experimental work is required to verify our account.

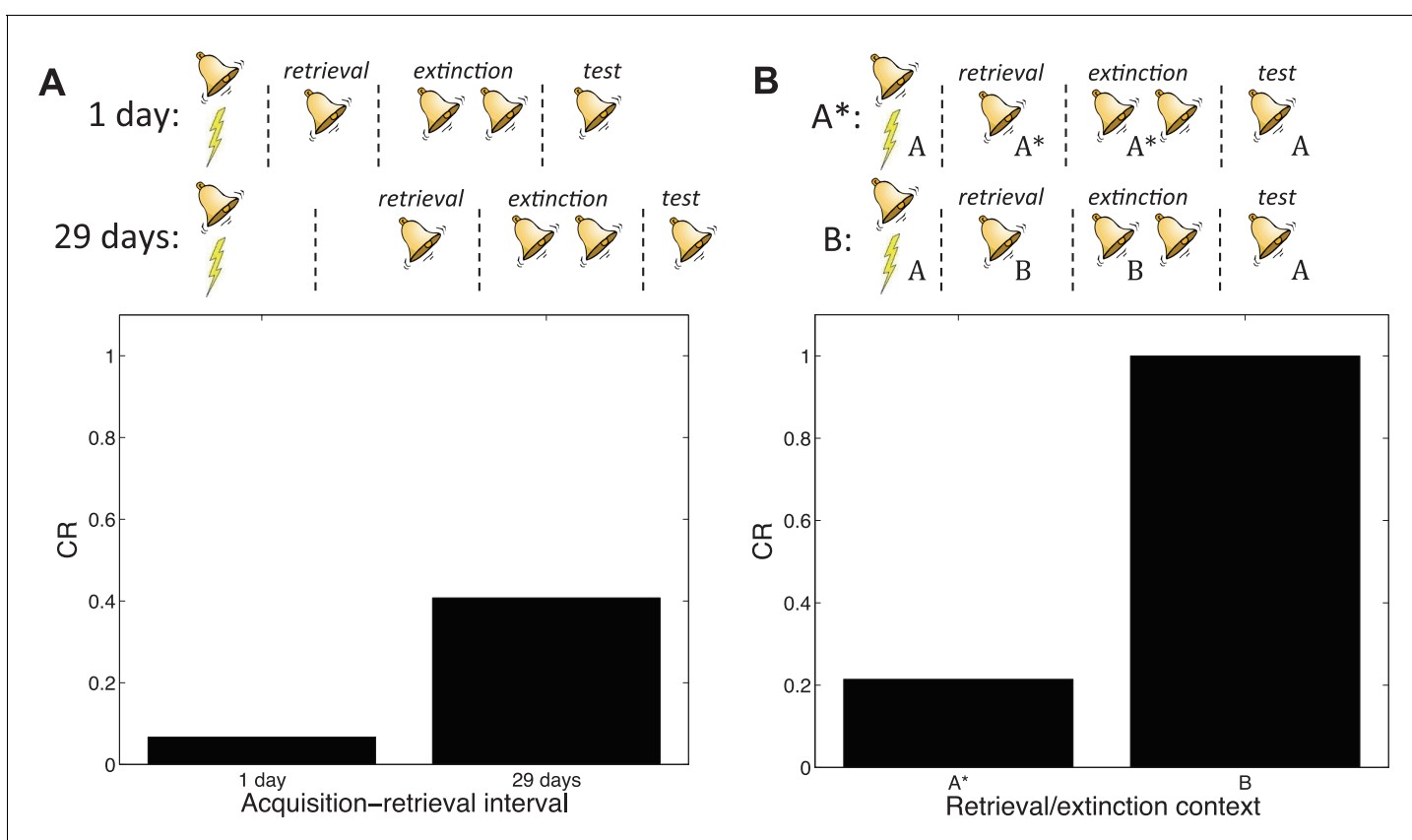

**Figure 14.** Boundary conditions in the Monfils-Schiller paradigm. (**A**) A short acquisition-retrieval interval is more effective at attenuating spontaneous recovery of fear at test than a long acquisition-retrieval interval. (**B**) A retrieval/extinction context (A*) that is similar to the acquisition context (**A**) leads to attenuated renewal of fear when tested in A, whereas a very dissimilar context (**B**) leads to renewal.

In summary, our theory provides a statistically principled explanation for the efficacy of the Monfils-Schiller paradigm in attenuating learned fear, and also reconciles some of the discrepant findings across different studies using this paradigm. These findings can therefore be regarded as delineating systematic boundary conditions on the original findings, although more work will be required to ascertain whether all the discrepancies identified by *Auber et al. (2013)* can be rationalized in this way.

### Paradoxical enhancement of memory

In the Monfils-Schiller paradigm, extinction follows memory retrieval; what happens if the extinction phase is omitted? It has been observed that retrieval without extinction leads to 'paradoxical' enhancement of a weak acquisition memory at the time of test (*Eysenck, 1968*; *Rohrbaugh and Riccio, 1970*; *Rohrbaugh et al., 1972*). Paradoxical enhancement of memory may be related to well-established 'testing effects', which show that memory is enhanced by the act of retrieval even in the absence of feedback (e.g., *Karpicke and Roediger, 2008*). This finding is paradoxical because no new learning has taken place, suggesting that retrieval enhances the retrievability of the memory (*Riccio et al., 2006*; *Spear, 1973*). There is also evidence that paradoxical enhancement is transient, disappearing with longer retrieval-test intervals (*Gisquet-Verrier and Alexinsky, 1990*).

Our model accounts for these findings (*Figure 15*) by positing that the retrieval trial induces the inference that the acquisition cause is once again active, and this inference persists into the test phase due to the contiguity principle. That is, when comparing to a test phase without the retrieval trial, the role of the retrieval trial in our theory is to 'bridge the gap' between training and test, and prolong the duration for which the acquisition cause is inferred to be active. The contiguity principle

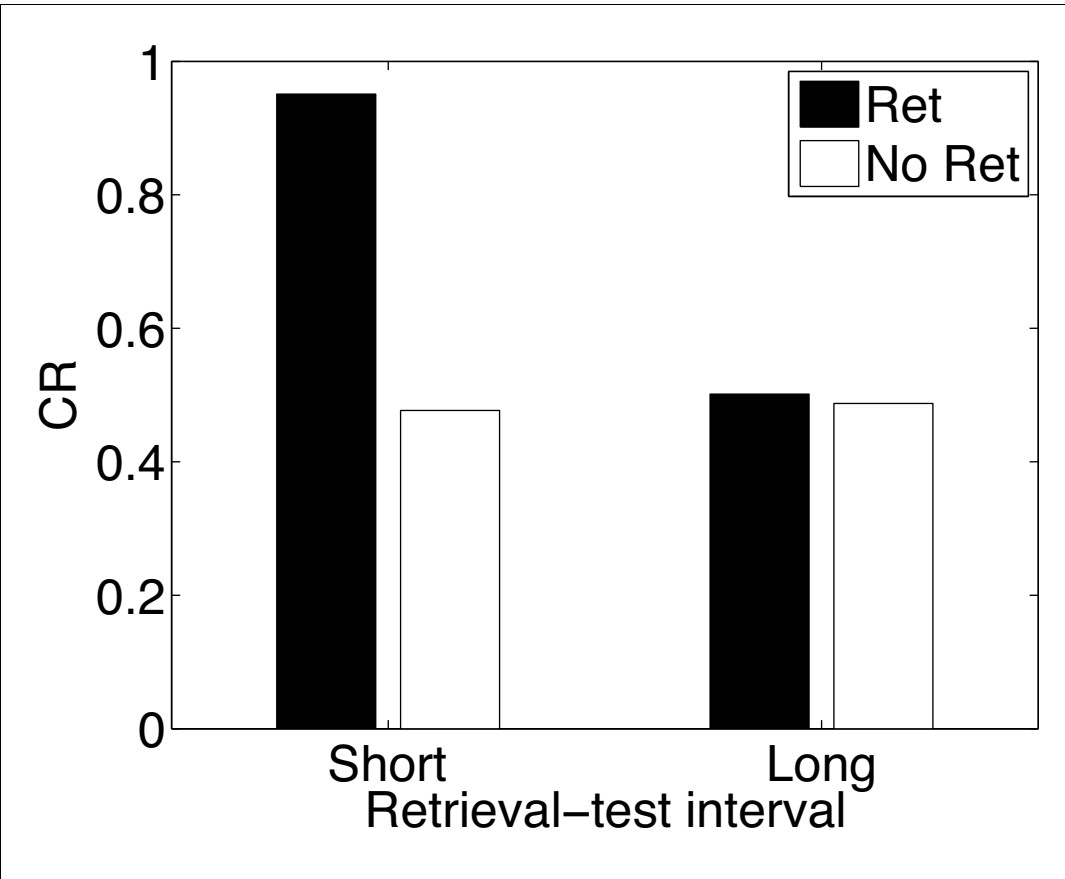

**Figure 15.** Paradoxical enhancement of memory. A weak conditioned response is first acquired using a low-intensity US (in order to prevent a ceiling effect). The graph shows the conditioned response at test following a retrieval cue (Ret) or no retrieval cue (No Ret) at short and long retrieval-test intervals.

also implies that the inference should be transient, which explains why longer retrieval-test intervals eliminate the effect. The paradoxical enhancement of memory due to retrieval can thus be explained by our model in terms of altered retrieval probability rather than new associative learning. Our model makes further predictions regarding the efficacy of the retrieval trial as a function of its delay after conditioning—delayed retrieval just before test should be less effective at enhancing fear at test, because it is less likely to reactivate the acquisition latent cause.

## Discussion

We have shown how major phenomena in post-retrieval memory modification can be accounted for by a rational analysis of Pavlovian conditioning. The key idea of our theory is a distinction between two learning processes: a structure learning process that infers the latent causes underlying sensory data, and an associative learning process that adjusts the parameters of the internal model so that each latent cause is likely to give rise to the sensory data ascribed to it. While this latter process has a statistical interpretation in our model, it is in practice similar to standard Rescorla-Wagner or reinforcement learning. The main difference is that our theory extends previous associative learning theories that concentrate on the latter process while assuming a fixed model of the environment, and formalizes the dynamic interplay between learning and memory, which we suggest is at the heart of post-retrieval memory modification phenomena. We further show that the theory can reproduce experimentally observed boundary conditions on memory modification, such as the effects of age, memory strength, and prediction error.

The discussion is organized as follows. First we discuss the main results of our theory, we then sketch a tentative neural implementation of our model. We then compare our model to previous models of the same phenomena, in particular one that is more tightly related to a neural implementation (*Osan et al., 2011*), and one originating in the human memory literature that has gained considerable empirical support (*Howard and Kahana, 2002*; *Howard et al., 2005*; *Sederberg et al., 2008*). Finally, we discuss the relationship between our framework and the popular notion of 'reconsolidation'—a hypothetical neural process by which previous memories can be modified upon retrieval.

### Explaining the mystery of the Monfils-Schiller paradigm

One of the most intriguing recent findings in the memory modification literature was the discovery of a noninvasive behavioral treatment that is effective at attenuating recovery of conditioned fear (*Agren et al., 2012*; *Monfils et al., 2009*; *Schiller et al., 2010*). Monfils, Schiller and their colleagues demonstrated (in both rodents and humans) that performing extinction training within a short interval following a retrieval cue (an unreinforced CS presentation) reduced later recovery of fear. The effect was later demonstrated in appetitive learning (*Ma et al., 2012*) and contextual fear conditioning (*Flavell et al., 2011*; *Rao-Ruiz et al., 2011*). The Monfils-Schiller paradigm has also been applied to drug-associated memory, attenuating drug-seeking in rats and cue-induced heroin craving in human addicts (*Xue et al., 2012*), as well as reducing cocaine-primed reinstatement of conditioned place preference (*Sartor and Aston-Jones, 2014*) and context-induced reinstatement of alcoholic beer seeking (*Millan et al., 2013*) in rats.

The Monfils-Schiller paradigm is theoretically tantalizing because it is not *a priori* clear what is the difference between the retrieval trial and the first trial of any extinction session—why is it that the CS-alone trial in the Monfils-Schiller paradigm acts as a 'retrieval cue', while the first CS-alone trial of a regular extinction session does not? Previous explanations had suggested that the retrieval cue starts a reconsolidation process, whereas the original (recalled) memory is rendered labile, and can be modified while it is being reconsolidated into long term memory. The idea was that the extinction session then modifies this labile memory, permanently rewriting it as a less fearful memory (*Monfils et al., 2009*). However, it is not clear why this should not happen in regular extinction, where the first extinction trial can also be seen as a retrieval cue that initiates a reconsolidation cascade. The effectiveness of this paradigm thus seems to challenge our basic understanding of the interplay between learning and memory processes.

Our theory resolves this puzzle by stressing the role of the extended period of learning (in our model, additional iterations of the EM algorithm) during the long retrieval-extinction gap, in which the rat is left in its home cage to 'ruminate' about its recent experience. Thus our explanation rests

not on the existence of a separate reconsolidation process that is invoked by the retrieval trial, but rather on the same learning and memory mechanisms that are at play in acquisition and extinction—the idea that inference about the latent structure of the environment affects whether new information will update an old association, or whether it will be attributed to a new memory (new latent cause). In this sense, according to our theory, the 'retrieval' trial is, in fact, not different from any other trial, and perhaps a more accurate nomenclature would be to call the retrieval-extinction interval an 'updating interval' rather than focus on a 'retrieval cue'.

Despite its successes, the effectiveness of the Monfils-Schiller paradigm has been controversial, with several replication failures (*Chan et al., 2010*; *Costanzi et al., 2011*; *Ishii et al., 2015*; *Kindt and Soeter, 2013*; *Ma et al., 2012*; *Soeter and Kindt, 2011*). *Auber et al. (2013)* described a number of methodological differences between these studies, possibly delineating boundary conditions on the Monfils-Schiller paradigm. Inspired by this suggestion, we showed through simulations that the consequences of several methodological differences (acquisition-retrieval interval and context similarity) are indeed predicted by our theory. Nevertheless, important boundary conditions on the length and characteristics of the retrieval-extinction interval remain to be studied; for instance, does it have to be longer than 10 min (as has been done in previous experiments) or is the minimum length of this gap more parametrically dependent on the overall pace of new information (e.g., the length of the ITIs at acquisition).

From a neurobiological standpoint, recent work has lent plausibility to the claim that the Monfils-Schiller paradigm erases the CS-US association learned during acquisition. After fear conditioning, there is an upregulation of AMPA receptor trafficking to the post-synaptic membrane at thalamus-amygdala synapses, and memory is impaired if this trafficking is blocked (*Rumpel et al., 2005*), suggesting that changes in post-synaptic AMPA receptor density may be the neural substrate of associative learning in fear conditioning. *Monfils et al. (2009)* reported increased phosphorylation of AMPA receptors in the lateral amygdala after the retrieval trial (a possible correlate of memory labilization), and also found that a second CS presented one hour after the first reversed the increase in AMPAr phosphorylation. *Clem and Huganir (2010)* found that extinction following retrieval resulted in synaptic removal of calcium-permeable AMPA receptors. The latter finding is significant in that it indicates a reversal of the synaptic changes that occurred during conditioning, supporting the view that the Monfils-Schiller paradigm results in unlearning of the original CS-US association. Furthermore, the Monfils-Schiller paradigm has been shown to induce neural modifications that are distinct from standard extinction (*Lee et al., 2015*; *Tedesco et al., 2014*).

Our theoretical analysis is consistent with these findings. We showed in simulations that during the retrieval-extinction interval, an associative learning process is engaged (and continues to be engaged during extinction training) that decrements the CS-US association, whereas in our model standard extinction engages a structure learning process that assigns the extinction trials to a new latent cause, creating a new memory trace without modifying the original memory.

## Neural implementation

Although we have so far not committed to any specific neural implementation of our model, we believe it fits comfortably into the computational functions of the circuit underlying Pavlovian conditioning. Here we propose a provisional mapping onto this circuit, centering on the amygdala and the 'hippocampal-VTA loop' (*Lisman and Grace, 2005*) connecting the hippocampus and the ventral tegmental area in the midbrain. Our basic proposal is inspired by two lines of research, one on the role of hippocampus in structure learning (*Aggleton et al., 2007*; *Gershman et al., 2010*, *2014*), and one on the role of the dopamine system and the amygdala (*Blair et al., 2001*) in associative learning.

In previous work, we have suggested that the hippocampus is a key brain region involved in partitioning streams of experience into latent causes (*Gershman et al., 2010*, *2014*). This view resonates with earlier models emphasizing the role of the hippocampus in encoding sensory inputs into a statistically compressed latent representation (*Fuhs and Touretzky, 2007*; *Gluck and Myers, 1993*; *Levy et al., 2005*). Some of the evidence for this view comes from studies showing that context-specific memories depend on the integrity of the hippocampus (e.g., *Honey and Good, 1993*), indicating that animals without a hippocampus cannot 'carve nature at its joints' (i.e., partition observations into latent causes; see *Gershman and Niv, 2010*; *Gershman et al., 2015*).

Within the current model, we propose that the dentate gyrus (DG) activates latent representations of the sensory inputs in area CA3. Each of these representations corresponds to a latent cause, and their level of activation is proportional to their prior probability (*Equation 3*). Mechanistically, these representations may be encoded in attractors by the dense recurrent collaterals that are characteristic of CA3 (*McNaughton and Morris, 1987*).

An important aspect of our model is that the repertoire of latent causes can expand adaptively. One potential mechanism for creating new attractors is neurogenesis of granule cells in the DG (*Becker, 2005*). This account predicts that the role of neurogenesis in creating new attractors should be time-sensitive in a manner comparable to the latent cause prior (i.e., it should implement the contiguity principle). Consistent with this hypothesis, *Aimone et al. (2006)* have suggested that immature granule cells, by virtue of their low activation thresholds, high resting potentials and constant turnover, cause inputs that are distant in time to map onto distinct CA3 representations. Furthermore, evidence suggests that new granule cells die over time if they are not involved in new learning (*Shors et al., 2012*), offering another mechanism by which the contiguity principle could be implemented.

Consistent with the idea that neurogenesis supports the partitioning of experience into latent causes, suppression of neurogenesis reduces both behavioral and neural discrimination between similar contexts (*Niibori et al., 2012*). Importantly, many CA3 neurons are temporally selective, responding to individual contexts only if exposures are separated by long temporal intervals, and this selectivity depends on intact neurogenesis (*Rangel et al., 2014*), as one would expect based on the contiguity principle. Our interpretation of neurogenesis predicts that its suppression (e.g., by irradiation of the DG), will force experiences separated by long temporal gaps to be assigned to the same latent cause, thus eliminating the age-based boundary condition on memory modification (*Alberini, 2007*; *Milekic and Alberini, 2002*; *Suzuki et al., 2004*).

There is widespread agreement that CS-US associations in auditory fear conditioning are encoded by synapses between the thalamus and the basolateral amygdala (BLA; *McNally et al., 2011*). Accordingly, we suggest that the amygdala transmits a US prediction that is then compared to sensory afferents from the periacqueductal gray region of the midbrain. The resultant prediction error is computed in the ventral tegmental area (VTA) and transmitted by dopaminergic projections to both the amygdala and CA1.

Our theory makes the testable prediction that disrupting the neural substrates of associative learning, or potentiating the neural substrates responsible for inferring new latent causes, during the retrieval-extinction interval should block memory updating in the Monfils-Schiller paradigm. Thus, both deactivating the BLA or stimulating the DG (e.g., using optogenetic manipulations) should block memory updating following retrieval. We also predict that the relative balance of activity in these two regions, measured for example using fMRI, should relate to individual differences in conditional responding in the test phase.

The role of dopamine in associative learning is well established (see *Glimcher, 2011* for a review), and has been specifically implicated in Pavlovian fear conditioning (*Pezze and Feldon, 2004*), although the role of dopamine in aversive conditioning is still a matter of controversy (*Mirenowicz and Schultz, 1996*; *Brooks and Berns, 2013*; *Cohen et al., 2012*). Dopamine gates synaptic plasticity in the basolateral amygdala (*Bissière et al., 2003*), consistent with its hypothesized role in driving the learning of CS-US associations. We hypothesize that dopaminergic inputs to CA1 have an additional role: influencing the posterior distribution over latent causes. That is, dopamine prediction errors can be used to assess the similarity of current sensory inputs to those expected by the current configuration of latent causes. Large discrepancies will cause the generation of a new latent cause, to account for the current unpredicted sensory input (see *Figure 3*). The output of CA1 further feeds back into the VTA by way of the subiculum (*Lisman and Grace, 2005*), potentially providing a mechanism by which the posterior distribution over latent causes can modulate the prediction errors, as suggested by our model. In appetitive conditioning experiments, (*Reichelt et al., 2013*) have shown that dysregulating dopaminergic activity in the VTA prevented the destabilization of memory by NMDA receptor antagonists (injected systemically following a retrieval trial), consistent with the hypothesis that dopaminergic prediction errors are necessary for memory updating after memory retrieval. It is not known whether this effect is mediated by dopaminergic projections to the hippocampus.

## Why expectation-maximization?

A key claim of this paper is that associative and structure learning are coupled: learning about associations depends on structural inferences, and vice versa. Our rational analysis suggested that this coupling can be resolved by alternating between the two forms of learning, using a form of the EM algorithm (*Dempster et al., 1977*; *Neal and Hinton, 1998*). While we do not believe that this is a literal description of the computational processes underlying learning, it is a useful abstraction for several reasons. First, EM is the standard method in machine learning for dealing with coupled problems of this form—namely, problems in which both latent variables and parameters are unknown. It is also closely related to variational inference algorithms (see *Neal and Hinton, 1998*), which have become a workhorse for scalable Bayesian computation. Second, variants of EM have become popular as theories of learning in the brain. For example, *Friston (2005)* suggests that it is a basic motif for synaptic plasticity in the cortex, and biologically plausible spiking neuron implementations have been put forth by *Deneve (2008)* and *Nessler et al. (2013)*. Third, as described in the Appendix, EM reduces to the Rescorla-Wagner model under particular parameter constraints. Thus, it is natural to view the model as a principled generalization of the most well-known account of Pavlovian conditioning. Fourth, the iterative nature of EM plays an important role in our explanation of the Monfils-Schiller effect: the balance between memory formation and modification shifts dynamically over multiple iterations, and we argued that this explains why a short period of quiescence prior to extinction training is crucial for observing the effect.

## Comparison with a mismatch-based autoassociative neural network

(*Osan et al., 2011*)have proposed an autoassociative neural network model of memory modification that explains many of the reported boundary conditions in terms of attractor dynamics (see *Amaral et al., 2008* also for a related model). In this model, acquisition and extinction memories correspond to attractors in the network, formed through Hebbian learning. Given a configuration of sensory inputs, the state of the network evolves towards one of these attractors. The retrieved attractor is then updated through Hebbian learning. In addition, a 'mismatch-induced degradation' process adjusts the associative weights that are responsible for the mismatch between the retrieved attractor and the current input pattern (i.e., the weights are adjusted to favor the input pattern). Mismatch is assumed to accumulate over the course of the input presentation.

The degradation process in this model can be viewed as a kind of error-driven learning: When the network does not accurately encode the current input, the weights are adjusted to encode it more accurately in the future. In the case of extinction, this implements a form of unlearning. The relative balance of Hebbian learning and mismatch-induced degradation determines the outcome of extinction training: assuming that the original shock pattern is retrieved at the beginning of extinction, degradation weakens the shock pattern, whereas Hebbian learning strengthens the retrieved shock pattern. Administration of PSIs is modeled as temporarily eliminating the influence of Hebbian plasticity on the weight update.

*Osan et al. (2011)* showed that their network model could account for a number of the boundary conditions on memory modification described above. For example, they simulated the effect of CS reexposure duration prior to PSI administration (*Eisenberg et al., 2003*; *Suzuki et al., 2004*) and suggested that post-reexposure PSI administration should have a tangible effect on the shock memory only for short, but not too short reexposure durations (i.e., what we modeled as 'short' duration in our simulations of the PSI experiments): for very short reexposure trials, the shock memory is preferentially retrieved because it has already been encoded in an attractor as a consequence of acquisition (i.e., the shock memory is the dominant trace). The accumulated mismatch is small, and hence mismatch-induced degradation has little effect on the shock memory. Since the mismatch is close to zero and the effect of PSIs is to turn off Hebbian learning, the net effect of PSI administration following reexposure is no change in the memory. On long reexposure trials, the accumulated mismatch becomes large enough to favor the formation of a new attractor corresponding to the extinction memory (i.e., the no-shock memory is the dominant trace). In this case, PSI administration will have little effect on the shock memory, because after a sufficiently long duration Hebbian learning is operating on a different attractor. Only in the case of intermediate-length reexposure, mismatch is large enough to induce degradation of the shock attractor, but not large enough to induce the formation of a new, no-shock attractor. The PSI prevents Hebbian learning from compensating for this

degradation by strengthening the shock attractor, so the result is a net decrease in the strength of the shock attractor.

In addition to the parametric effect of reexposure duration on reconsolidation, *Osan et al. (2011)* also simulated the effects of memory strength (more highly trained memories are resistant to disruption by PSI administration), the effects of NMDA receptor agonists (which have the opposite effects of PSIs), and the effects of blocking mismatch-induced degradation (the amnestic effect of PSI administration is attenuated). However, the model of *Osan et al. (2011)* is fundamentally limited by the fact that it lacks an explicit representation of time within and between trials. This prevents it from accounting for the results of the Monfils-Schiller paradigm: all the retrieval-extinction intervals should lead to the same behavior (contrary to the empirical data). The lack of temporal representation also prevents it from modeling the effects of memory age on reconsolidation, since there is no mechanism for taking into account the interval between acquisition and reexposure. In contrast, our model explicitly represents temporal distance between observations, making it sensitive to changes in timing.[12] Conceivably, one could incorporate a time-sensitive mechanism into the Osan model by using a 'temporal context' signal that drifts slowly over time (see *Sederberg et al., 2011*).

Another, related issue with the model of *Osan et al. (2011)* is that in order to explain spontaneous recovery, it was necessary to introduce an ad hoc function that governs pattern drift during reexposure. This function—by construction—produces spontaneous recovery, but it is not obvious why pattern drift should follow such a function, and no psychological or neurobiological justification was provided. Nonetheless, an appealing feature of the *Osan et al. (2011)* model is its neurobiological plausibility. We know that attractor networks exist in the brain (e.g., in area CA3 of the hippocampus), and (in certain circumstances) support the kinds of learning described above. The model is appealing as it provides a simplified but plausible mapping from computational variables to biological substrates.

As we discussed in the previous section, one way to think about latent causes at a neural level is in terms of attractors (e.g., in area CA3). Thus, although the formal details of *Osan et al. (2011)* differ from our own, there may be neural implementations of the latent cause model that bring it closer to the formalism of the attractor network. However, in its current form, our model is not specified at the same biologically detailed level as the model of *Osan et al. (2011)*; our model makes no distinction between Hebbian plasticity and mismatch-induced degradation, and consequently has nothing to say about pharmacological manipulations that selectively affect one or the other process, for example the disruption of mismatch-induced degradation by inhibitors of the ubiquitin-proteasome cascade (*Lee et al., 2008*).

## Comparison with stimulus sampling and retrieved context models

One of the first formal accounts of spontaneous recovery from extinction was developed by *Estes, (1955)*. In his *stimulus sampling theory*, the nominal stimulus is represented by a collection of stimulus elements that change gradually and randomly over time. These stimulus elements enter into association with the US, such that the CR is proportional to the number of conditioned elements. When the CS is presented again at a later time, the CR it elicits will thus depend on the overlap between its current vector of stimulus elements and the vector that was present during conditioning. Extinction reverses the learning process, inactivating the currently active conditioned elements. However, some conditioned elements will not be inactive during the extinction phase (due to stimulus sampling). As the interval between extinction and test increases, these elements will randomly reenter the stimulus representation, thereby producing spontaneous recovery of the extinguished CR. This theory has since been elaborated in a number of significant ways to accommodate a wide variety of memory phenomena (*Howard, 2014*).

While stimulus sampling theory, on the surface, appears quite different from our latent cause theory, there are some intriguing connections. The assumption that the same nominal stimulus can have different representations at different times is central to both accounts. Our theory posits latent stimulus elements (causes) that change over time, but these elements are not directly observable by the animal; rather, the structure learning system constructs a representation of these elements through Bayesian inference. Knowledge about gradual change is built into the prior through the contiguity principle. Like stimulus sampling theory, our theory views spontaneous recovery as a consequence of the extinction memory's waning through random fluctuation. Again, this fluctuation is inferred rather than observed.

The structure learning system acquires explicit distributional information about the latent causes—information that is absent from the stimulus sampling theory as developed by *Estes (1955)*. As a consequence, in our framework the representation of a stimulus contains information about its history and the history of other stimuli that were inferred to have been generated by the same latent cause. Because of the contiguity principle, stimuli that occur nearby in time are likely to have been generated by the same latent cause; this means that the 'temporal context' of a stimulus figures prominently in the distributional information stored by the structure learning system.

The Temporal Context Model (TCM; *Howard et al., 2005*; *Howard and Kahana, 2002*; *Sederberg et al., 2008*) can be viewed as a modern-day elaboration of the Estes stimulus-sampling model; rather than relying on random drift, it maintains a gradually changing 'context vector' of recent stimulus history that gets bound to stimulus vectors through Hebbian learning. The context vector can be used to cue retrieval of stimuli from memory (as in free recall tasks), which in turn causes the reinstatement of context bound to the retrieved stimuli. One way to view our latent cause theory is as a particular rationalization of retrieved context models like TCM: the 'context' representation corresponds to the posterior over latent causes, retrieving context corresponds to inferring a latent cause, and updating the stimulus-context associations corresponds to updating the sufficient statistics of the posterior (i.e., structure learning). Indeed, precisely this correspondence was made by *Socher et al. (2009)*, where a latent cause model of text corpora was used as the underlying internal model for word lists.

The latent cause model extends TCM by positing additional constraints on context drift. For example, in the latent cause model, the diagnosticity of sensory observations matters: a sensory observation that is highly diagnostic of a change in latent causes could have a very large effect on the posterior probabilities that the agent assigns to latent causes (and thus its 'context', if we consider latent causes to be coextensive with context). TCM in its original form does not incorporate any notion of diagnosticity—it merely computes a running average of sensory observations and retrieved contextual information. Bayesian versions of TCM, such as the one developed by *Socher et al. (2009)*, could potentially capture effects of diagnosticity, although such effects have not yet been systematically investigated.

Connecting our theoretical work with retrieved context models like TCM allows us to make contact with a relevant segment of the human episodic memory literature studying post-retrieval memory modification (*Chan et al., 2009*; *Chan and LaPaglia, 2013*; *Forcato et al., 2007*, *2010*; *Hupbach et al., 2007*, *2009*). In one line of research developed by Hupbach and colleagues, the researchers used a list-learning paradigm to show that reminding participants of one list (A) shortly before asking them to study a second list (B) produced an asymmetric pattern of intrusions at test: participants intruded a large number of items from list B when asked to recall list A, but not vice versa (*Hupbach et al., 2007*). When no reminder was given, participants showed an overall low level of intrusions across list A and list B recall.

One interpretation of these findings, in line with reconsolidation accounts of memory modification, is that the reminder caused the memory of list A to become labile, thereby allowing list B items to become incorporated into the list A memory. However, *Sederberg et al. (2011)* showed that the findings of Hupbach and colleagues could be accounted for by TCM (see also *Gershman et al., 2013c* for converging neural evidence), further suggesting that retrieved context models are relevant to understanding post-retrieval memory modification, but more work is needed to flesh out the correspondences sketched here. Briefly, a latent cause theory might be able to account for the Hupbach results if one assumes that a latent cause associated with list A is retrieved at at the beginning of list B (analogous to the retrieval of the list A temporal context).

## Consolidation and reconsolidation

In developing our theory of memory modification, we have studiously avoided the term 'reconsolidation' that appears ubiquitously throughout the literature we have modeled. Reconsolidation, like many concepts in the study of learning, has a dual meaning as both a set of empirical phenomena and as a theoretical hypothesis about the nature of those phenomena (*Rudy et al., 2006*). The theoretical hypothesis is derived from the idea that newly formed memories are initially labile (sensitive to disruption or modification), but over time undergo a protein synthesis-dependent 'consolidation' process that converts them into a stable molecular format largely resistant to disruption (*McGaugh, 1966*, *2000*). Here we are specifically discussing 'synaptic consolidation' that unfolds

over seconds to minutes, in contrast to the 'systems consolidation' that unfolds over days to months and is hypothesized to involve the transfer of memory from hippocampus to neocortex (*Dudai, 2012*). The discovery that post-retrieval PSI administration was effective at disrupting memory long outside the consolidation window (*Nader et al., 2000*) inspired the idea that memory retrieval renders memory once again labile, requiring a second phase of consolidation (named 'reconsolidation') to stabilize the memory. Like initial consolidation, reconsolidation requires protein synthesis, explaining why PSIs disrupt memory stabilization.

We have avoided this terminology for several reasons. First, our theory is formulated at a level of abstraction that does not require commitment to a particular model of synaptic consolidation or reconsolidation. The process by which a memory becomes progressively resistant to disruption can be modeled in various ways (e.g., *Fusi et al., 2005*; *Clopath et al., 2008*; *Ziegler et al., 2015*), and it is currently unclear to what extent these biological mechanisms are consistent with normative models of learning (see *Gershman, 2014,* for one attempt at connecting the levels of analysis *Gershman, 2014*). In particular, our theory does not incorporate an explicit consolidation process; increased resistance to disruption as a function of time arises from the contiguity principle, which implies that beliefs about a latent cause are less likely to be modified by new experience if a long interval has elapsed since the latent cause was believed to be active. Similarly, we do not explicitly model reconsolidation; post-retrieval lability arises from the increased probability that an old latent cause is active once again.

A second reason we have avoided the consolidation/reconsolidation terminology is that the underlying theoretical claims face longstanding difficulties. Most theories of synaptic consolidation assume that amnestic agents like PSIs degrade the memory engram, and the post-learning (or post-retrieval) consolidation window reflects a period of time during which the trace is vulnerable to degradation. However, as a number of authors have pointed out (*Miller and Springer, 1973*, *2006*; *Lewis, 1979*), amnesia could alternatively arise through disrupted memory retrieval. In other words, the amnestic agent might make a memory harder to retrieve, while sparing the engram. This retrieval-oriented view is consistent with the observation that pre-test reminders (e.g., the US or training context) can cause recovery from amnesia (*Lewis et al., 1968b*; *Miller and Springer, 1972*; *Quartermain et al., 1972*). Another difficulty facing consolidation theory is that the putative consolidation window could be reduced to less than 500 msec (far shorter than the hypothesized speed of synaptic consolidation) if animals were familiarized with the learning environment (*Lewis et al., 1968a*, *1969*; *Miller, 1970*).

These difficulties inspired a family of retrieval-oriented theories that contrast starkly with storage-oriented consolidation theories (see *Miller and Matzel, 2006*; *Riccio et al., 2006* for recent reviews). In an influential paper, *Lewis et al. (1968b)* argued that experimental amnesia results from the impairment of a retrieval pathway rendered labile by reminders. Importantly, this impairment is temporary: a sufficiently salient reminder can activate the impaired retrieval pathway or possibly establish a new retrieval pathway. This idea is compatible with the stimulus sampling framework described in the previous section, where retrieval cues both activate prior memory traces and contribute new stimulus elements to the trace. Another retrieval-oriented theory, advocated by Riccio and colleagues (*Millin et al., 2001*; *Riccio et al., 2006*), views experimental amnesia as a state-dependent retrieval impairment. Specifically, the animal's physiological state is a powerful retrieval cue, so by testing animals in the absence of the amnestic agent (hence in a different physiological state), typical experimental amnesia experiments induce an encoding-retrieval mismatch (*Spear, 1973*; *Tulving and Thomson, 1973*). This idea lead to the counterintuitive prediction, subsequently confirmed, that administration of amnestic agents prior to test would reinstate the impaired memory (*Gisquet-Verrier et al., 2015*; *Hinderliter et al., 1975*).

The difficulties facing consolidation theory do not necessarily pose problems for our theory, and indeed we showed that our theory predicts the transience of experimental amnesia as well as the reminder effect of pre-test PSI administration. Nonetheless, we see merit in both encoding-oriented and retrieval-oriented theories, since our theory asserts critical roles for both encoding and retrieval processes. In our simulations, we have shown that manipulations can affect both the strength of the CS-US association and also the probability of retrieving that association.

## Conclusion

One challenge to developing a unified theory of memory modification is that some of the basic facts are still disputed. Some authors have found that Pavlovian contextual fear memories become labile

after retrieval (*Debiec et al., 2002*), while others have not (*Biedenkapp and Rudy, 2004*), and yet others argue that memory modification is transient (*Frankland et al., 2006*; *Power et al., 2006*). A similar situation exists for instrumental memories: some studies have shown that instrumental memories undergo post-retrieval modification (*Fuchs et al., 2009*; *Milton et al., 2008*), while others have not (*Hernandez and Kelley, 2004*). The literature on post-retrieval modification of human procedural memories has also been recently thrown into doubt (*Hardwicke et al., 2016*). There are many differences between these studies that could account for such discrepancies, including the type of amnestic agent, how the amnestic agent is administered (systemically or locally), the type of reinforcer, and the timing of stimuli. Despite these ambiguities, we have described a number of regularities in the literature and how they can be accounted for by a latent cause theory of conditioning. The theory offers a unifying normative account of memory modification that links learning and memory from first principles.

## Materials and methods

In this section, we provide the mathematical and implementational details of our model. Code is available at *Gershman, 2017*, https://github.com/sjgershm/memory-modification (with a copy archived at https://github.com/elifesciences-publications/memory-modification).

### The expectation-maximization algorithm

The EM algorithm, first introduced by *Dempster et al. (1977)*, is a method for performing maximum-likelihood parameter estimation in latent variable models. In our model, the latent variables correspond to the vector of latent cause assignments, $\mathbf{z}_{1:t}$, the parameters correspond to the associative weights, $\mathbf{W}$, and the data correspond to the CS-US history, $\mathcal{D}_{1:t} = \{\mathbf{X}_{1:t}, \mathbf{r}_{1:t}\}$, where $\mathbf{X}_{1:t} = \{\mathbf{x}_1, \dots, \mathbf{x}_t\}$ and $\mathbf{r}_{1:t} = \{r_1, \dots, r_t\}$. Let $Q(\mathbf{z}_{1:t})$ be a distribution over $\mathbf{z}_{1:t}$. The EM algorithm can be understood as performing coordinate ascent on the functional

$$\begin{aligned}\mathcal{F}(\mathbf{W}, Q) &= \sum_{\mathbf{z}_{1:t}} Q(\mathbf{z}_{1:t} | \mathcal{D}_{1:t}) \log P(\mathbf{z}_{1:t}, \mathcal{D}_{1:t} | \mathbf{W}) \\ &= \sum_{\mathbf{z}_{1:t}} Q(\mathbf{z}_{1:t} | \mathcal{D}_{1:t}) \log[P(\mathcal{D}_{1:t} | \mathbf{z}_{1:t}, \mathbf{W}) P(\mathbf{z}_{1:t})].\end{aligned}$$

By Jensen's inequality, this functional is a lower bound on the log marginal likelihood of the data, $\log P(\mathcal{D}_{1:t} | \mathbf{W}) = \log \sum_{\mathbf{z}_{1:t}} P(\mathcal{D}_{1:t}, \mathbf{z}_{1:t} | \mathbf{W})$, which means that maximizing $\mathcal{F}$ corresponds to optimizing the internal model to best predict the observed data (*Neal and Hinton, 1998*).

The EM algorithm alternates between maximizing $\mathcal{F}(\mathbf{W}, Q)$ with respect to $\mathbf{W}$ and $Q$. Letting $n$ indicate the iteration,

$$\text{E-step}: Q^{n+1} \leftarrow \underset{Q}{\text{argmax}}\ \mathcal{F}(\mathbf{W}^n, Q)$$
$$\text{M-step}: \mathbf{W}^{n+1} \leftarrow \underset{\mathbf{W}}{\text{argmax}}\ \mathcal{F}(\mathbf{W}, Q^{n+1})$$

Alternating the E and M steps repeatedly, $\mathcal{F}(\mathbf{W}, Q)$ is guaranteed to converge to a local maximum (*Neal and Hinton, 1998*). It can also be shown that $\mathcal{F}(\mathbf{W}, Q)$ is maximized with respect to $Q(\mathbf{z}_{1:t})$ when $Q = P(\mathbf{z}_{1:t} | \mathcal{D}_{1:t}, \mathbf{W})$. Thus, the optimal E-step is exact Bayesian inference over the latent variables $\mathbf{z}_{1:t}$.

There are two challenges facing a biologically and psychologically plausible implementation of this algorithm. First, the E-step is intractable, since it requires summing over an exponentially large number of possible latent cause assignments. Second, both steps involve computations operating on the entire history of observations, whereas a more plausible algorithm is one that operates online, one observation at a time (*Anderson, 1990*). Below we summarize an approximate, online form of the algorithm. To reduce notational clutter, we drop the $n$ superscript (indicating EM iteration), and implicitly condition on $\mathbf{W}$.

### The E-step: structure learning

The E-step corresponds to calculating the posterior using Bayes' rule:

$$q_{tk} = P(z_t = k | \mathcal{D}_{1:t}) = \frac{\sum_{\mathbf{z}_{1:t-1}} P(\mathcal{D}_t | z_t = k, \mathcal{D}_{1:t-1}) P(z_t = k | \mathbf{z}_{1:t-1})}{\sum_j \sum_{\mathbf{z}_{1:t-1}} P(\mathcal{D}_t | z_t = j, \mathcal{D}_{1:t-1}) P(z_t = j | \mathbf{z}_{1:t-1})}. \tag{13}$$

Note that the number of terms in the summation over $\mathbf{z}_{1:t-1}$ grows exponentially over time; consequently, calculating the posterior exactly is intractable. Following (*Anderson, 1991*), we use a 'local' *maximum a posteriori* (MAP) approximation see for more discussion (*Sanborn et al., 2010*):

$$q_{tk} \approx \frac{P(\mathcal{D}_t | z_t = k, \hat{\mathbf{z}}_{1:t-1}, \mathcal{D}_{1:t-1}) P(z_t = k | \hat{\mathbf{z}}_{1:t-1})}{\sum_j P(\mathcal{D}_t | z_t = j, \hat{\mathbf{z}}_{1:t-1}, \mathcal{D}_{1:t-1}) P(z_t = j | \hat{\mathbf{z}}_{1:t-1})}, \tag{14}$$

where $\hat{\mathbf{z}}_{1:t-1}$ is defined recursively according to:

$$\hat{z}_t = \underset{k}{\operatorname{argmax}} \, P(\mathcal{D}_t | z_t = k, \hat{\mathbf{z}}_{1:t-1}, \mathcal{D}_{1:t-1}) P(z_t = k | \hat{\mathbf{z}}_{1:t-1}). \tag{15}$$

In other words, the local MAP approximation is obtained by replacing the summation over partitions with the sequence of conditionally optimal cluster assignments. Although this is not guaranteed to arrive at the globally optimal partition (i.e., the partition maximizing the posterior over all time-points), in our simulations it tends to produce very similar solutions to more elaborate approximations like particle filtering (*Gershman and Niv, 2010*; *Sanborn et al., 2010*). The local MAP approximation has also been investigated in the statistical literature. *Wang and Dunson (2011)* found that it compares favorably to fully Bayesian inference, while being substantially faster.

The first term in *Equation 15* (the likelihood) is derived using standard results in Bayesian statistics (*Bishop, 2006*):

$$P(\mathcal{D}_t | z_t = k, \hat{\mathbf{z}}_{1:t-1}, \mathcal{D}_{1:t-1}) = \mathcal{N}(r_t; \hat{r}_{tk}, \sigma_r^2) \prod_{d=1}^{D} \mathcal{N}(x_{td}; \hat{x}_{tkd}, v_{tk}^2), \tag{16}$$

where

$$\hat{r}_{tk} = \sum_{d=1}^{D} x_{td} w_{kd} \tag{17}$$

$$\hat{x}_{tkd} = \frac{N_{tk} \bar{x}_{tkd}}{N_{tk} + \sigma_x^2} \tag{18}$$

$$v_{tk}^2 = \frac{\sigma_x^2}{N_{tk} + \sigma_x^2} + \sigma_x^2. \tag{19}$$

Here $N_{tk}$ denotes the number of times $z_\tau = k$ for $\tau < t$ and $\bar{x}_{tkd}$ denotes the average cue values for observations assigned to cause $k$ for $\tau < t$. The second term in *Equation 15* (the prior) is given by the time-sensitive Chinese restaurant process (*Equation 3*).

## The M-step: sssociative learning

The M-step is derived by differentiating $\mathcal{F}$ with respect to $\mathbf{W}$ and then taking a gradient step to increase the lower bound. This corresponds to a form of stochastic gradient ascent, and is in fact remarkably similar to the Rescorla-Wagner learning rule (see below). Its main departure lies in the way it allows the weights to be modulated by a potentially infinite set of latent causes. Because these latent causes are unknown, the animal represents an approximate distribution over causes, $\mathbf{q}$ (computed in the E-step). The components of the gradient are given by:

$$[\nabla \mathcal{F}]_{kd} = \sigma_r^{-2} x_{td} \delta_{tk}, \tag{20}$$

where $\delta_{tk}$ is given by *Equation 7*. To make the similarity to the Rescorla-Wagner model clearer, we absorb the $\sigma_r^{-2}$ factor into the learning rate, $\eta$.

## Simulation parameters

With two exceptions, we used the following parameter values in all the simulations: $\alpha = 0.1, \eta = 0.3, \sigma_r^2 = 0.4, \sigma_x^2 = 1, \theta = 0.02, \lambda = 0.01$. For modeling the retrieval-extinction data, we treated $\theta$ and $\lambda$ as free parameters, which we fit using least-squares. For simulations of the human

data in *Figure 13*, we used $\theta = 0.0016$ and $\lambda = 0.00008$. Note that $\theta$ and $\lambda$ change only the scaling of the predictions, not their direction; all ordinal relationships are preserved.

The CS was modeled as a unit impulse: $x_{td} = 1$ when the CS is present and 0 otherwise (similarly for the US). Intervals of 24 hr were modeled as 20 time units; intervals of one month were modeled as 200 time units. While the choice of time unit was somewhat arbitrary, our results do not depend strongly on these particular values.

## Relationship to the Rescorla-Wagner model

In this section we demonstrate a formal correspondence between the classic Rescorla-Wagner model and our model. In the Rescorla-Wagner model, the outcome prediction $\hat{r}_t$ is, as in our model, parameterized by a linear combinations of the cues $\mathbf{x}_t$ and is updated according to the prediction error:

$$\hat{r}_t = \sum_{d=1}^{D} w_d x_{td} \tag{21}$$

$$\delta_t = r_t - \hat{r}_t \tag{22}$$

$$\mathbf{w} \leftarrow \mathbf{w} + \eta \mathbf{x}_t \delta_t. \tag{23}$$

The key difference is that in our model, we allow there to be separate weight vectors for each latent cause. When $\alpha = 0$, the distribution over latent causes reduces to a delta function at a single cause (since the probability of inferring new latent causes is always 0), and hence there is only a single weight vector. In this case, the two models coincide.

## Acknowledgements

This research was supported by a Graduate Research Fellowship from the National Science Foundation (SJG), a Sloan Research Fellowship (YN), and NIMH grants R01MH091147, R21MH086805 (MHM). The authors thank Daniela Schiller and Marc Howard for helpful comments.

## Additional information

### Funding

| Funder | Grant reference number | Author |
| --- | --- | --- |
| National Science Foundation | Graduate research fellowship | Samuel J Gershman |
| Sloan Research Foundation | Sloan Research Fellowship | Yael Niv |
| National Institutes of Health | R01MH091147 | Marie-H Monfils |
| National Institutes of Health | R21MH086805 | Marie-H Monfils |

The funders had no role in study design, data collection and interpretation, or the decision to submit the work for publication.

### Author contributions

SJG, Conceptualization, Software, Formal analysis, Visualization, Methodology, Writing—original draft, Writing—review and editing; M-HM, KAN, YN, Conceptualization, Writing—original draft, Writing—review and editing

### Author ORCIDs

Samuel J Gershman, http://orcid.org/0000-0002-6546-3298

Yael Niv, http://orcid.org/0000-0002-0259-8371

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
