## [Decision Letter]

Thank you for submitting your article "The computational nature of memory modification" for consideration by *eLife*. Your article has been reviewed by two peer reviewers, Marc Howard (Reviewer #1) and Brandon Turner (Reviewer #2), and the evaluation has been overseen by Michael Frank as the Reviewing Editor and Richard Ivry as the Senior Editor.

The reviewers have discussed the reviews with one another and the Reviewing Editor has drafted this decision to help you prepare a revised submission.

Summary:

The present article develops and extensively tests a new computational theory of memory modification. Here, memory traces are modified based on a structural learning mechanism involving the inferred latent causes. The simulations they present show that the model can account for many of the experimental effects while holding a set of model parameters constant (for the most part). The idea of latent causes that are autocorrelated in time, and that modulate associations is an intriguing hypothesis that has the potential to make sense of a broad range of behavioral phenomena, and reconsolidation has been a topic of much recent interest.

Essential revisions:

1) All involved were largely enthusiastic about the contribution, but there was some concern about the relevance of the work to the biological sciences (neuroscience). One reviewer was concerned that there wasn't enough links, noting that there was no modeling of neural data or a serious mapping between the components of the model and neural circuits (even though some ideas along these lines are presented in the Discussion), and that it is not clear how the model would be implemented by biological neurons. The other reviewer noted that the section on neural implementation was highly speculative and could potentially be removed to cut down on space. It would be helpful to clarify the extent to which you think your work interfaces with the neurosciences and if appropriate, further emphasize the link and describe predictions relevant for neural manipulations and/or interpretation of recordings etc., if there are indeed clear predictions.

2) We thought it would be helpful to establish a closer connection to empirical data. You do an excellent job in reviewing the basic experimental effects, but this feels very abstract. Given that so much emphasis is directed at the relative differences in "CR" (isn't this P(CR)?) across conditions, it would be great to know what strengths of each effect should be expected. (This is not a model fitting expedition, so we are not expecting impressive model fits or anything, but some guidelines for the magnitude of the effects would be helpful.) For example, you could base a statistic of these other studies cited, something simple like probability of CR, overlaid with data and model, to ensure that the model is at least on par with the data. The issue is that you relate the simulations to data via statements like “high recovery of fear was observed in a test on the following day”, whereas you could just calculate what the recovery was (i.e., a probability) and use this as a guide in interpreting say, Figure 5.

3) One reviewer noted that the first few pages of the Introduction assumed too much prior knowledge of the reader (especially for a general journal). Could you elaborate on the basic details of the experimental paradigms? As someone who is unfamiliar with this literature, it was hard to follow the description of the experimental effects because the experiment itself had not been described. For example, section 1.1 reads beautifully and sets up the subsequent discussion nicely, but the first few pages seem to be predicated on this later section.

4) It is unclear why the EM algorithm is used here. It seems like an overly complicated assumption that really isn't justified well. You cite this Friston (2005) article, but can something more be done to explain why this choice was made (as opposed to others?)

5) Maybe some code could be provided to produce some of the simulations? The models seem pretty easy to set up, but it might be better for dissemination purposes to offer up a link to a simple simulation of the model.

6) It would be helpful to know more about the model parameters, and specifically what the model can predict and cannot predict. It was stated in the manuscript that the parameters were chosen 'heuristically' but that the basic patterns were observed for many other values of the parameters. Can more be said within the manuscript about which combination of model parameters fail to produce the desired effects? The critical question is whether the predictions from the model are consistent with the data because of its architecture itself, or is it just one specific version of the model that happens to work?

---

## [Author Response]

*Essential revisions:*

*1) All involved were largely enthusiastic about the contribution, but there was some concern about the relevance of the work to the biological sciences (neuroscience). One reviewer was concerned that there wasn't enough links, noting that there was no modeling of neural data or a serious mapping between the components of the model and neural circuits (even though some ideas along these lines are presented in the Discussion), and that it is not clear how the model would be implemented by biological neurons. The other reviewer noted that the section on neural implementation was highly speculative and could potentially be removed to cut down on space. It would be helpful to clarify the extent to which you think your work interfaces with the neurosciences and if appropriate, further emphasize the link and describe predictions relevant for neural manipulations and/or interpretation of recordings etc., if there are indeed clear predictions.*

Thank you for this suggestion. We think that making a more direct link with biology is very important. Our approach, following a Marr-style analysis, is to first understand the design principles and algorithmic structure of the system before contemplating its neural implementation. This point is now clarified in the beginning of the Introduction:

“It is important to clarify at the outset that our theory is formulated at an abstract, cognitive level of analysis, in order to elucidate the design principles and algorithmic structure of memory. We do not make strong claims about biologically plausible implementation in realistic neurons, although we speculate about such an implementation in the Discussion. Addressing this question is a logical next step for this line of research.”

This analysis alone took up 70+ pages, so we feel that a thorough treatment of the implementation issue deserves its own paper. Nonetheless, we agree that more links can be made in the present paper. We have addressed this by providing new neural predictions in several places throughout the manuscript:

“These simulations suggest empirically testable predictions. For example, our earlier work on context-dependent learning argued that young animals and animals with hippocampal lesions have small values of α (Gershman, Blei & Niv, 2010). […] Thus, we expect that pharmacological manipulations and individual variation of these receptors should be systematically related to post-retrieval memory modification.”

“Our interpretation of neurogenesis predicts that its suppression (e.g., by irradiation of the DG), will force experiences separated by long temporal gaps to be assigned to the same latent cause, thus eliminating the age-based boundary condition on memory modification (Alberini, 2007; Milkecic & Alberini, 2002; Suzuki et al., 2004).”

“Our theory makes the testable prediction that disrupting the neural substrates of associative learning, or potentiating the substrates of structure learning, during the retrieval-extinction interval should block memory updating in the Monfils-Schiller paradigm. […] We also predict that the relative balance of activity in these two regions, measured for example using fMRI, should relate to individual differences in conditional responding in the test phase.”

*2) We thought it would be helpful to establish a closer connection to empirical data. You do an excellent job in reviewing the basic experimental effects, but this feels very abstract. Given that so much emphasis is directed at the relative differences in "CR" (isn't this P(CR)?) across conditions, it would be great to know what strengths of each effect should be expected. (This is not a model fitting expedition, so we are not expecting impressive model fits or anything, but some guidelines for the magnitude of the effects would be helpful.) For example, you could base a statistic of these other studies cited, something simple like probability of CR, overlaid with data and model, to ensure that the model is at least on par with the data. The issue is that you relate the simulations to data via statements like “high recovery of fear was observed in a test on the following day”, whereas you could just calculate what the recovery was (i.e., a probability) and use this as a guide in interpreting say, Figure 5.*

This is a useful suggestion, although in practice it is difficult because the various phenomena we model are distilled from a heterogeneous collection of experimental paradigms. Some level of abstraction from the original data was necessary in order to avoid having to develop specific linking assumptions for different paradigms, which we felt would have distracted from our main points. As a compromise, we have added data from two studies to Figure 4, in order for the reader to see how the ordinal empirical results correspond to the corresponding ordinal model results. We have also added the following clarification in a footnote:

“Note that we have not fit any parameters to the empirical data, because this would require specifying many different linking assumptions in order to accommodate the diverse range of paradigms under consideration. Instead, we have attempted to reproduce the findings qualitatively using the same parameters for all simulations.”

With regard to P(CR) vs. CR: this is typically a continuous measure (e.g., proportion freezing in a Pavlovian fear conditioning experiment). In our model, we use the threshold formulation as a continuous response measure, rather than a probability of a binary response.

*3) One reviewer noted that the first few pages of the Introduction assumed too much prior knowledge of the reader (especially for a general journal). Could you elaborate on the basic details of the experimental paradigms? As someone who is unfamiliar with this literature, it was hard to follow the description of the experimental effects because the experiment itself had not been described. For example, section 1.1 reads beautifully and sets up the subsequent discussion nicely, but the first few pages seem to be predicated on this later section.*

Thank you for pointing this out. We have reorganized and expanded the Introduction so that the basic details of a Pavlovian conditioning experiment are described up-front:

“During the acquisition phase of a typical Pavlovian conditioning experiment, a motivationally neutral conditional stimulus (CS; e.g., tone) is repeatedly paired with a motivationally reinforcing unconditional stimulus (US; e.g., a shock). […] A final test phase, after some delay, probes the animal's long-term memory of the CS-US relationship by presenting the CS alone.”

*4) It is unclear why the EM algorithm is used here. It seems like an overly complicated assumption that really isn't justified well. You cite this Friston (2005) article, but can something more be done to explain why this choice was made (as opposed to others?)*

We agree that this choice needs further justification. We have added a section to the Discussion (“Why expectation-maximization?”):

“A key claim of this paper is that associative and structure learning are coupled: learning about associations depends on structural inferences, and vice versa. […] Fourth, the iterative nature of EM plays an important role in our explanation of the Monfils-Schiller effect: the balance between memory formation and modification shifts dynamically over multiple iterations, and we argued that this explains why a short period of quiescence prior to extinction training is crucial for observing the effect.”

*5) Maybe some code could be provided to produce some of the simulations? The models seem pretty easy to set up, but it might be better for dissemination purposes to offer up a link to a simple simulation of the model.*

Agreed. We have created a github repository with the code: https://github.com/sjgershm/memory-modification

We reference code availability in Materials and methods.

*6) It would be helpful to know more about the model parameters, and specifically what the model can predict and cannot predict. It was stated in the manuscript that the parameters were chosen 'heuristically' but that the basic patterns were observed for many other values of the parameters. Can more be said within the manuscript about which combination of model parameters fail to produce the desired effects? The critical question is whether the predictions from the model are consistent with the data because of its architecture itself, or is it just one specific version of the model that happens to work?*

To address this point, we have added new simulations (Figure 5) showing

parameter-dependence, along with additional discussion and experimental predictions:

*“*As discussed in the previous section, these effects are parameter-dependent. The key parameters of interest are the concentration parameter α and the US variance σ^2^, which jointly determine sensitivity to prediction errors. […] Thus, we expect that pharmacological manipulations and individual variation of these receptors should be systematically related to post-retrieval memory modification.*”*

We also added a pointer to these simulations, where we discuss the influence of different parameters on memory modification: “Parameter-dependence is examined systematically in the next section.”